# PAC-Bayesian Spectrally-Normalized Bounds for Adversarially Robust Generalization

**Jiancong Xiao**[1,*], **Ruoyu Sun**[2,3,4,†], **Zhi-Quan Luo**[2,4,†]
[1]University of Pennsylvania, PA, USA
[2]Scool of Data Science, The Chinese University of Hong Kong, Shenzhen, China
[3]Shenzhen International Center for Industrial and Applied Mathematcs
[4]Shenzhen Research Institute of Big Data
jcxiao@upenn.edu, {sunruoyu,luozq}@cuhk.edu.cn

## Abstract

Deep neural networks (DNNs) are vulnerable to adversarial attacks. It is found empirically that adversarially robust generalization is crucial in establishing defense algorithms against adversarial attacks. Therefore, it is interesting to study the theoretical guarantee of robust generalization. This paper focuses on norm-based complexity, based on a PAC-Bayes approach (Neyshabur et al., 2017b). The main challenge lies in extending the key ingredient, which is a weight perturbation bound in standard settings, to the robust settings. Existing attempts heavily rely on additional strong assumptions, leading to loose bounds. In this paper, we address this issue and provide a spectrally-normalized robust generalization bound for DNNs. Compared to existing bounds, our bound offers two significant advantages: Firstly, it does not depend on additional assumptions. Secondly, it is considerably tighter, aligning with the bounds of standard generalization. Therefore, our result provides a different perspective on understanding robust generalization: The mismatch terms between standard and robust generalization bounds shown in previous studies do not contribute to the poor robust generalization. Instead, these disparities solely due to mathematical issues. Finally, we extend the main result to adversarial robustness against general non-$\ell_p$ attacks and other neural network architectures.

## 1 Introduction

Even though deep neural networks (DNNs) have impressive performance on many machine learning tasks, they are often highly susceptible to adversarial perturbations imperceptible to the human eye (Goodfellow et al., 2015; Madry et al., 2018). They have received enormous attention in the machine learning literature over recent years and a large number of defense algorithms (Gowal et al., 2020; Rebuffi et al., 2021) are proposed to improve the robustness in practice. Nonetheless, it still fails to deliver satisfactory performance. One major challenge stems from adversarially robust generalization. For example, Madry et al. (2018) demonstrated that the robust generalization gap can extend up to 50% on CIFAR-10. In contrast, the standard generalization gap is notably small in practical settings. Hence, a theoretical question arises: Why is there a huge difference between standard generalization and robust generalization? This paper focuses on norm-based generalization analysis.

In classical learning theory, one of the most well-known findings is that the generalization bound for neural networks depends on the norms of their layers (Bartlett, 1998). To further explore the generalization of deep learning, a series of work aimed at improving the norm-based bound (Bartlett &

---

[*]Work done while at CUHK(SZ).

[†]Corresponding Authors.

37th Conference on Neural Information Processing Systems (NeurIPS 2023).

Mendelson, 2002; Neyshabur et al., 2015; Golowich et al., 2018), mainly using tools of Rademacher complexity. The tightest bound is given by Bartlett et al. (2017), using a covering number approach. Neyshabur et al. (2017b) gave a different and simpler proof based on PAC-Bayes analysis, presented an almost equally tight bound. The key step involves bounding the change in output of the predictors in response to slight variations in the predictor parameters. In particular, considering $f_{\mathbf{w}}(\mathbf{x})$ as the predictor parameterized by $\mathbf{w}$, the crucial component for providing the generalization bound lies in bounding the gap $|f_{\mathbf{w}}(\mathbf{x}) - f_{\mathbf{w}'}(\mathbf{x})|$, where $\mathbf{w}$ and $\mathbf{w}'$ are close. The weight perturbation bound, which addresses this aspect, is presented in Lemma 2 of Neyshabur et al. (2017b).

To comprehend the limited robust generalization capabilities of deep learning, a line of research endeavors to extend the norm-based bounds into robust settings. However, this has proven to be a challenging mathematical problem, as researchers have attempted the mentioned approaches including the Rademacher complexity (Khim & Loh, 2018; Yin et al., 2019; Awasthi et al., 2020), covering number (Gao & Wang, 2021; Xiao et al., 2022a; Mustafa et al., 2022), and the PAC-Bayes analysis (Farnia et al., 2018), yet a satisfactory solution remains elusive. For more details, see Section 2.

We use the PAC-Bayesian approach as an example to illustrate the mathematical challenge. The weight perturbations in adversarial settings differ from those in standard settings. When considering two predictors $f_{\mathbf{w}}(\cdot)$ and $f_{\mathbf{w}'}(\cdot)$, the adversarial examples against these predictors are distinct, leading to a gap referred to as robust weight perturbation (defined later in Problem 1). It remains unclear how to establish a bound for robust weight perturbation. The combined changes in input and weights can potentially cause a significant alteration in the function value. The main challenge is illustrated in Figure 1, the details of which will be provided in Section 6.2. As a result, Farnia et al. (2018) introduced additional assumption to control this

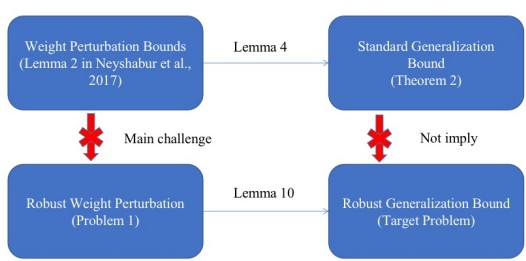

Figure 1: Demonstration of the main challenge of providing robust generalization bound. The weight perturbation bound (Neyshabur et al., 2017b) seems hard to extend to adversarial settings.

gap and provide bounds in adversarial settings. However, the assumption imposed limitations on the effectiveness of the bounds due to two reasons: Firstly, the assumption of sharp gradients throughout the domain is a strong requirement. Secondly, without this assumption, the bounds become unbounded ($=+\infty$). Similarly, other existing norm-based bounds also depend on additional assumptions or involve higher-order terms in certain factors.

Given that the existing robust generalization bounds are much larger than standard generalization bounds, these results suggest a possible hypothesis: The significant disparity between standard and robust generalization in practical scenarios could potentially be attributed to the mismatch terms between the standard bounds and the robust bounds. However, verifying this hypothesis is challenging because it remains unclear whether the existence of these terms or assumptions is due to mathematical issues. Therefore, the current bounds are insufficient to address the main theoretical question.

In this paper, we address this problem and present a PAC-Bayes spectrally-normalized robust generalization bound without additional assumptions. Our robust generalization bound is as tight as the standard generalization bound, with an additional factor representing the perturbation intensity $\epsilon$. Furthermore, our bound is strictly smaller than the previous generalization bounds proposed in adversarial robustness settings. To provide an initial overview of the main result, we begin by defining the *spectral complexity* of a $d$-layer neural network $f_{\mathbf{w}}$ as follows:

$$\Phi(f_{\mathbf{w}}) = \Pi_{i=1}^{d} \|W_i\|_2^2 \sum_{i=1}^{d} (\|W_i\|_F^2 / \|W_i\|_2^2), \tag{1}$$

where $W_i$ is the weights of $f_{\mathbf{w}}$ in each of the $d$ layers.

**Theorem** (Informal). *Let $m$ be the number of samples and the training samples $x$ is bounded by $B$. $\epsilon$ is the attack intensity. Let $f_{\mathbf{w}} : \mathcal{X} \to \mathbb{R}^k$ be a $d$-layer feedforward network. Then, with high probability, we have*

$$Robust\ Generalization \leq \mathcal{O}(\sqrt{(B+\epsilon)^2 \Phi(f_{\mathbf{w}})/m}).$$

When $\epsilon = 0$, the bound reduces to the standard generalization bound presented by Neyshabur et al. (2017b). Our results give a different perspecitve from existing bounds. The additional factors or assumptions are solely due to mathematical considerations. Our findings suggest that the implicit difference of the spectral complexity $\Phi(f_\mathbf{w})$ likely contributes to the significant disparity between standard and robust generalization.

**Technical Proof.** It is shown that the robust weight perturbation is not controllable without additional assumptions. Therefore, existing tools are not sufficient to derive the bounds. The main technical tools to derive the bounds are two folds. Firstly, we introduce a crucial inequality to address this problem, which is the preservation of weight perturbation bound under $\ell_p$ attack. Secondly, we restructure the proof by (Neyshabur et al., 2017b) in terms of the margin operator. This modification enables the application of the aforementioned inequality. To further extend the bound to more general settings, we establish a framework that allows us to derive a robust generalization bound from its corresponding standard generalization bound. The

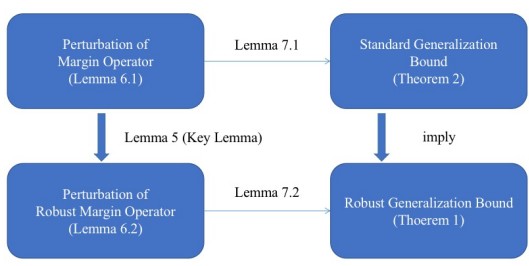

Figure 2: Demonstration of the framework: perturbation bound of robustified function. Under this framework, a standard generalization bound directly implies a robust generalization bound.

framework's demonstration is presented in Figure 2, and detailed information regarding Figure 2 will be provided in Section 6.3.

Furthermore, we extend the results to encompass general settings. Firstly, although $\ell_p$ adversarial attacks are widely used, real-world attacks are not always bounded by the $\ell_p$ norm. Hence, we extend the results to cover general attacks. Secondly, as the current state-of-the-art robust performance is achieved with WideResNet (Rebuffi et al., 2021; Croce et al., 2021), we demonstrate that the results can be extended to other DNN structures, such as ResNet.

The contributions are listed as follows:

1. Main result: We provide a PAC-Bayesian spectrally-normalized robust generalization bound without any additional assumption. The derived bound is as tight as the standard generalization bound and tighter than the existing robust generalization bound.

2. Our results give a different perspecitve from existing bounds. The significant disparity between standard and robust generalization in practical scenarios is not attributed to the mismatch terms between the standard bound and the robust bound. The implicit difference of the spectral complexity $\Phi(f_\mathbf{w})$ possibly contributes to the significant disparity.

3. We provide a general framework for robust generalization analysis. We show how to obtain a robust generalization bound from a given standard generalization bound.

4. We extend the result to general adversarial attacks and other neural networks architectures.

## 2    Related Work

**Adversarial Attack.** Adversarial examples were first introduced in (Szegedy et al., 2014). Since then, adversarial attacks have received enormous attention (Papernot et al., 2016; Moosavi-Dezfooli et al., 2016; Carlini & Wagner, 2017). Nowadays, attack algorithms have become sophisticated and powerful. For example, Autoattack (Croce & Hein, 2020) and Adaptive attack (Tramer et al., 2020). Therefore, we consider theoretical analysis on robust margin loss (defined later in Eq. (4)) against any norm-based attacks. Real-world attacks are not always norm-bounded (Kurakin et al., 2018). Therefore, we also consider non-$\ell_p$ attacks (Lin et al., 2020; Xiao et al., 2022c) in Sec. 7.

**Adversarially Robust Generalization.** Even enormous algorithms were proposed to improve the robustness of DNNs (Madry et al., 2018; Tramèr et al., 2018; Gowal et al., 2020; Rebuffi et al., 2021), the performance was far from satisfactory. One major issue is the poor robust generalization, or robust overfitting (Rice et al., 2020). A series of studies (Xing et al., 2021; Xiao et al., 2022b,d; Ozdaglar et al., 2022) have delved into the concept of uniform stability within the context of adversarial training.

However, these analyses focused on general Lipschitz functions, without specific consideration for neural networks.

**Rademacher Complexity.** Rademacher complexity can provide similar spectral norm generalization bound as PAC-Bayesian bound (Theorem 2). Rademacher complexity was extended to adversarial settings for linear classifier (Khim & Loh, 2018; Yin et al., 2019) and two-layers neural networks (Awasthi et al., 2020). As for DNNs, they found that it was mathematically difficult and provided some discussions on surrogate losses rather than the adversarial loss.

**Covering Number.** Rademacher complexity can be bounded in terms of the covering number of the function class, as discussed in (Bartlett et al., 2017). Nevertheless, calculating the covering number for an adversarial function class is also shown to be a challenging problem. Gao & Wang (2021) considered adversarial loss against FGSM attacks, employing similar assumptions to those of (Farnia et al., 2018), resulting in a bound similar to Theorem 3. Additionally, Xiao et al. (2022a) and Mustafa et al. (2022) introduced two different methods, respectively, to compute the covering number for adversarial function classes. However, the bounds obtained through these methods remain notably larger when compared to those in standard settings. The related research on Rademacher complexity and covering number help proves the difficulty of the problem we are addressing.

**PAC-Bayes Analysis.** We mainly compare our results to the previous PAC-Bayesian spectrally-normalized bounds (Neyshabur et al., 2017b; Farnia et al., 2018), which we have already discussed in the introduction. We will provide more details later. The workshop version of this paper is presented in (Xiao et al., 2023). Other PAC-Bayes frameworks for tackling adversarial robustness also exist. Viallard et al. (2021) explored a distinct adversarial attack targeting the loss of the $Q$-weighted majority vote over the posterior distribution $Q$. Mustafa et al. (2023) introduced a non-vacuous PAC-Bayes bound designed for stochastic neural networks.

## 3 Preliminaries

### 3.1 Notations

We mainly follow the notations of (Neyshabur et al., 2017b). Consider the classification task that maps the input $\mathbf{x} \in \mathcal{X}$ to the label $y \in \mathbb{R}^k$. The output of the model is a score for each of the $k$ classes. The class with the maximum score will be the prediction of the label of $\mathbf{x}$. A sample dataset $S = \{(\mathbf{x}_1, y_1), \cdots, (\mathbf{x}_m, y_m)\}$ with $m$ training samples is given. The $l_2$ norm of each of the samples $x_i$ is bounded by $B$, *i.e.*, $\|x_i\|_2 \leq B$, $i = 1, \cdots, m$. Let $\|W\|_F$ and $\|W\|_2$ denote the Frobenius norm and the spectral norm of the weights $W$, respectively.

**Fully-Connected Neural Networks.** Let $f_{\mathbf{w}}(\mathbf{x}) : \mathcal{X} \rightarrow \mathbb{R}^k$ be the function computed by a $d$-layer feed-forward network for the classification task with parameters $\mathbf{w} = \text{vec}\left(\{W_i\}_{i=1}^d\right)$, $f_{\mathbf{w}}(\mathbf{x}) = W_d\,\phi(W_{d-1}\,\phi(....\phi(W_1\mathbf{x})))$, here $\phi$ is the ReLU activation function. Let $f_{\mathbf{w}}^i(\mathbf{x})$ denote the output of layer $i$ before activation and $h$ be an upper bound on the number of output units in each layer. We can then define fully-connected feed-forward networks recursively: $f_{\mathbf{w}}^1(\mathbf{x}) = W_1\mathbf{x}$ and $f_{\mathbf{w}}^i(\mathbf{x}) = W_i\phi(f_{\mathbf{w}}^{i-1}(\mathbf{x}))$. In Section 7, we extend the results to ResNet (He et al., 2016), since the state-of-the-art robust performance is built on WideResNet (Rebuffi et al., 2021; Croce et al., 2021).

### 3.2 Standard Margin Loss and Robust Margin Loss

**Standard Margin Loss.** For any distribution $\mathcal{D}$ and margin $\gamma > 0$, the expected margin loss is defined as follows:

$$L_\gamma(f_{\mathbf{w}}) = \mathbb{P}_{(\mathbf{x},y)\sim\mathcal{D}} \left[ f_{\mathbf{w}}(\mathbf{x})[y] \leq \gamma + \max_{j\neq y} f_{\mathbf{w}}(\mathbf{x})[j] \right]. \tag{2}$$

Let $\widehat{L}_\gamma(f_{\mathbf{w}})$ be the empirical estimate of the above expected margin loss. Since setting $\gamma = 0$ corresponds to the classification loss, we will use $L_0(f_{\mathbf{w}})$ and $\widehat{L}_0(f_{\mathbf{w}})$ to refer to the expected loss and the training loss. The loss $L_\gamma$ defined this way is bounded between 0 and 1.

**Robust Margin Loss.** Adversarial examples are usually crafted by an attack algorithm. Let $\delta_{\mathbf{w}}^{adv}(\mathbf{x})$ be an algorithm output and $\delta_{\mathbf{w}}^*(\mathbf{x})$ be the maximizer of the following maximization problem

$$\max_{\|\delta\|\leq\epsilon} \ell(f_{\mathbf{w}}(\mathbf{x} + \delta), y), \tag{3}$$

where $\ell$ is the loss function of the predicted label and true label. Without explicit specification, $\|\cdot\|$ refers to the $\ell_2$ norm. The robust margin loss is defined as follows:

$$
\begin{aligned}
R_\gamma(f_\mathbf{w}) =& \mathbb{P}_{(\mathbf{x},y)\sim\mathcal{D}}\left[\exists \mathbf{x}' \in \mathbb{B}^p_\mathbf{x}(\epsilon), f_\mathbf{w}(\mathbf{x}')[y] \leq \gamma + \max_{j\neq y} f_\mathbf{w}(\mathbf{x}')[j]\right] \\
=& \mathbb{P}_{(\mathbf{x},y)\sim\mathcal{D}}\left[f_\mathbf{w}(\mathbf{x}+\delta^*_\mathbf{w}(\mathbf{x}))[y] \leq \gamma + \max_{j\neq y} f_\mathbf{w}(\mathbf{x}+\delta^*_\mathbf{w}(\mathbf{x}))[j]\right].
\end{aligned}
\tag{4}
$$

Let $\hat{R}_\gamma(f_\mathbf{w})$ be the empirical estimate of the above expected robust margin loss. The robust margin loss requires the whole norm ball around the original example $\mathbf{x}$ to be labelled correctly, which is the goal of norm-based adversarial robustness. By replacing $\delta^*_\mathbf{w}(\mathbf{x})$ by $\delta^{adv}_\mathbf{w}(\mathbf{x})$ in the above definition, we denote $R^{adv}_\gamma(f_\mathbf{w})$ as the margin loss against attacks $adv$. The work of (Farnia et al., 2018) consider three attacks: fast gradient sign method (FGSM or FGM), projected gradient method (PGM), and wasserstein risk minimization (WRM), *i.e.*, $adv$ = FGSM, PGM, and WRM. They provided three different bounds for these adversarial attacks respectively. However, methods for generating these adversarial examples are becoming significantly more sophisticated and powerful. For example, Autoattack (Croce & Hein, 2020) in default settings is a collection of four attacks to find adversarial examples. Therefore, a bound of robust margin loss against a single attack provides a limited robustness guarantee to a machine learning model. In fact, Autoattack collects different attacks to attempt and to provide a close lower estimation of $R_0(f_\mathbf{w})$. Therefore, this paper focuses on the robust margin loss.

## 4 Robust Generalization Bound

In this section, we will first provide our main result of robust generalization.

**Theorem 1** (**Main Result:** Robust Generalization Bound)**.** *For any $B, d, h, \epsilon > 0$, let $f_\mathbf{w} : \mathcal{X} \to \mathbb{R}^k$ be a $d$-layer feedforward network with ReLU activations. Then, for any $\delta, \gamma > 0$, with probability $\geq 1 - \delta$ over a training set of size $m$, for any $\mathbf{w}$, we have:*

$$
R_0(f_\mathbf{w}) - \hat{R}_\gamma(f_\mathbf{w}) \leq \mathcal{O}\left(\sqrt{\frac{(B+\epsilon)^2 d^2 h \ln(dh)\Phi(f_\mathbf{w}) + \ln\frac{dm}{\delta}}{\gamma^2 m}}\right),
$$

*where $\Phi(f_\mathbf{w}) = \Pi^d_{i=1}\|W_i\|^2_2 \sum^d_{i=1}\frac{\|W_i\|^2_F}{\|W_i\|^2_2}$ is the **spectral complexity** of $f_\mathbf{w}$.*

**Remark.** Theorem 1 is presented under $\ell_2$ attacks to simplify the notation. For other $\ell_p$ attacks, suppose all the samples $x_i$ have $\ell_p$ norm bounded by $B$ and $\|\delta\|_p \leq \epsilon$, the robust generalization bound is to replace $(B+\epsilon)$ by $\max\{1, n^{\frac{1}{2}-\frac{1}{p}}\}(B+\epsilon)$ in Theorem 1, where $n$ is the dimension of the samples $x_i$.

Theorem 1 provides the first PAC-Bayesian bound in adversarial robustness settings without introducing new assumptions. Fixing other factors, the generalization gap goes to 0 as $m \to \infty$.

**Theorem 2** (Standard Generalization Bound (Neyshabur et al., 2017b))**.** *For any $B, d, h > 0$, let $f_\mathbf{w} : \mathcal{X} \to \mathbb{R}^k$ be a $d$-layer feedforward network with ReLU activations. Then, for any $\delta, \gamma > 0$, with probability $\geq 1 - \delta$ over a training set of size $m$, for any $\mathbf{w}$, we have:*

$$
L_0(f_\mathbf{w}) - \widehat{L}_\gamma(f_\mathbf{w}) \leq \mathcal{O}\left(\sqrt{\frac{B^2 d^2 h \ln(dh)\Phi(f_\mathbf{w}) + \ln\frac{dm}{\delta}}{\gamma^2 m}}\right),
$$

*where $\Phi(f_\mathbf{w}) = \Pi^d_{i=1}\|W_i\|^2_2 \sum^d_{i=1}\frac{\|W_i\|^2_F}{\|W_i\|^2_2}$.*

**Comparison with Existing Standard Generalization Bounds.** Comparing the robust generalization bound in Theorem 1 with the standard generalization bound in Theorem 2, the only difference is a factor of the attack intensity $\epsilon$, which is unavoidable in adversarial settings. In other words. $B$ and $B + \epsilon$ are the magnitudes of the clean and adversarial examples, respectively. Therefore, our main result is as tight as the standard generalization bound in Theorem 2.

**Theorem 3** (Robust Generalization Bound (Farnia et al., 2018))**.** *For any $B, d, h > 0$, let $f_{\mathbf{w}} : \mathcal{X} \to \mathbb{R}^k$ be a $d$-layer feedforward network with ReLU activations. Consider an FGM attack with noise power $\epsilon$ according to Euclidean norm $\| \cdot \|_2$. Assume that $\|\nabla_{\mathbf{x}}\ell(f_{\mathbf{w}}(\mathbf{x}), y)\| \geq \kappa$, $\forall \mathbf{x}$ $\epsilon$-close to $\mathcal{X}$. Then, for any $\delta, \gamma > 0$, with probability $\geq 1 - \delta$ over a training set of size $m$, for any $\mathbf{w}$, we have:*

$$R_0^{adv}(f_{\mathbf{w}}) - \hat{R}_\gamma^{adv}(f_{\mathbf{w}}) \leq \mathcal{O}\left(\sqrt{\frac{(B+\epsilon)^2 d^2 h \ln(dh) \Phi^{fgm}(f_{\mathbf{w}}) + \ln\frac{dm}{\delta}}{\gamma^2 m}}\right),$$

$$\Phi^{fgm}(f_{\mathbf{w}}) = \prod_{i=1}^{d} \|W_i\|_2^2 \left(1 + C^{fgm}\right) \sum_{i=1}^{d} \frac{\|W_i\|_F^2}{\|W_i\|_2^2}, \text{ and } C^{fgm} = \frac{\epsilon}{\kappa}(\prod_{i=1}^{d} \|W_i\|_2)(\sum_{i=1}^{d} \prod_{j=1}^{i} \|W_j\|_2).$$

**Remark:** For robust generalization bounds of PGM or WRM adversarial attacks, the bounds have similar forms as in Theorem 3, with different constants $C^{pgm}$ and $C^{wrm}$.

**Comparison with Existing Robust Generalization Bounds.** Comparing Theorem 1 and Theorem 3, the difference of the upper bounds is the difference of $\Phi$ and $\Phi^{fgm}$, where $\Phi^{fgm}$ contains an additional term $C^{fgm}$. Therefore, our bound is tighter. Moreover, the robust generalization gap is much larger than the FGSM generalization gap based on the observation in practice. We provide a tighter upper bound for a larger generalization gap.

Additionally, the term $C^{fgm}$ could be very large. Notice that Theorem 3 requires $\ell(f_{\mathbf{w}}(\mathbf{x}), y)$ to be sharp w.r.t. $\mathbf{x}$ for all $\mathbf{x} \in \mathcal{X}$. It is hard to verify and $\kappa$ could be small. Therefore, if we remove the additional assumption $\|\nabla_{\mathbf{x}}\ell(f_{\mathbf{w}}(\mathbf{x}), y)\| \geq \kappa$, we have $C^{fgm} \to +\infty$ as $\kappa \to 0$ and the upper bound in Theorem 3 goes to infinity.

It is also worth noting that our bound is tighter than other norm-based robust generalization bounds derived in Rademacher complexity and covering number approaches, since these bounds are larger than their standard counterpart, the bound given by (Bartlett et al., 2017).

# 5 Analysis of Adversarially Robust Generalization

As mentioned in the introduction, the robust generalization gap is much larger than the standard generalization gap in practical scenarios. What factors contribute to such a significant difference? Previous norm-based bounds might lead to the following hypothesis: The significant disparity could potentially be attributed to the additional terms or assumptions between the standard bound and the robust bound. Our result provides a different perspective: They are solely due to mathematical considerations. The following three factors are (implicitly) different in Theorem 1 and Theorem 2 and possibly contribute to the significant disparity.

**Clean Sample and Adversarial Example ($B$ and $B + \epsilon$).** The only difference between the bounds in Theorem 1 and Theorem 2 lies in the factor $\epsilon$. In this context, $B$ represents the magnitude of clean samples, while $B + \epsilon$ signifies the magnitude of adversarial examples. This factor holds less significance in improving robust generalization, as it is unlikely to be controlled during the training of DNNs.

**Standard Margin and Robust Margin ($\gamma$).** The margin $\gamma$ remains consistent in both of these two bounds, but it is implicitly different in the definitions of standard margin loss and robust margin loss. The robust margin is smaller due to the smaller distance between two adversarial examples. As it is discussed in (Neyshabur et al., 2017a), $\gamma$ is usually considered to normalize the spectral complexity discussed below.

**Standard-Trained and Adversarially-Trained Parameters ($\Phi(f_{\mathbf{w}})$).** The spectral complexity $\Phi(f_{\mathbf{w}})$ is implicitly different because the weights $\mathbf{w}$ of the standard-trained and adversarially-trained models are distinct. The spectral complexity $\Phi(f_{\mathbf{w}})$ induced by adversarial training is significantly larger. We conducted experiments training MNIST, CIFAR-10, and CIFAR-100 datasets on VGG networks, see Appendix C. See also the work of (Xiao et al., 2022a) for more discussion about the experiments of weights norm of adversarially-trained models. The margin-normalized spectral complexity $\Phi(f_{\mathbf{w}})$ likely contributes to the huge difference between standard generalization and robust generalization.

# 6 Main Challenge of Robust Generalization Bound and Proof Sketch

## 6.1 PAC-Bayesian Framework

The PAC-Bayesian framework (McAllester, 1999) provides generalization guarantees for randomized predictors drawn from a learned distribution $Q$ (as opposed to a single predictor) that depends on the training data set. In particular, let $f_{\mathbf{w}}$ be a predictor parameterized by $\mathbf{w}$. We consider the distribution $Q$ over predictors of the form $f_{\mathbf{w}+\mathbf{u}}$, where $\mathbf{u}$ is a random variable and $\mathbf{w}$ is considered to be fixed. Given a prior distribution $P$ over the set of predictors that is independent of the training data, the PAC-Bayes theorem states that with probability at least $1 - \delta$, the expected loss of $f_{\mathbf{w}+\mathbf{u}}$ can be bounded as follows

$$\mathbb{E}_{\mathbf{u}}[L_0(f_{\mathbf{w}+\mathbf{u}})] \leq \mathbb{E}_{\mathbf{u}}[\widehat{L}_0(f_{\mathbf{w}+\mathbf{u}})] + 2\sqrt{\frac{2\left(KL\left(\mathbf{w}+\mathbf{u}\|P\right) + \ln\frac{2m}{\delta}\right)}{m-1}}. \tag{5}$$

To get a bound on the margin loss $L_0(f_{\mathbf{w}})$ for a single predictor $f_{\mathbf{w}}$, we need to relate the expected loss, $\mathbb{E}_{\mathbf{u}}[L_0(f_{\mathbf{w}+\mathbf{u}})]$ over a distribution $Q$, with the loss $L_0(f_{\mathbf{w}})$ for a single model. The following lemma provides this relation.

**Lemma 4** (Neyshabur et al. (2017b)). *Let $f_{\mathbf{w}}(\mathbf{x}) : \mathcal{X} \to \mathbb{R}^k$ be any predictor (not necessarily a neural network) with parameters $\mathbf{w}$, and $P$ be any distribution on the parameters that is independent of the training data. Then, for any $\gamma, \delta > 0$, with probability $\geq 1 - \delta$ over the training set of size $m$, for any $\mathbf{w}$, and any random perturbation $\mathbf{u}$ s.t. $\mathbb{P}_{\mathbf{u}}\left[\max_{\mathbf{x} \in \mathcal{X}} |f_{\mathbf{w}+\mathbf{u}}(\mathbf{x}) - f_{\mathbf{w}}(\mathbf{x})|_{\infty} < \frac{\gamma}{4}\right] \geq \frac{1}{2}$, we have:*

$$L_0(f_{\mathbf{w}}) \leq \widehat{L}_{\gamma}(f_{\mathbf{w}}) + 4\sqrt{\frac{KL\left(\mathbf{w}+\mathbf{u}\|P\right) + \ln\frac{6m}{\delta}}{m-1}}.$$

As it is discussed in (Neyshabur et al., 2017a), the KL-divergence is evaluated for a fixed $\mathbf{w}$ and $\mathbf{u}$ is random. Lemma 4 is not specific to neural networks and generally holds for any functions. Providing Lemma 4, it is left to provide a bound of $\|f_{\mathbf{w}+\mathbf{u}}(\mathbf{x}) - f_{\mathbf{w}}(\mathbf{x})\|_2$ to obtain the final generalization bound.[3] This framework can be directly extended to adversarially robust settings by replacing $\|f_{\mathbf{w}+\mathbf{u}}(\mathbf{x}) - f_{\mathbf{w}}(\mathbf{x})\|_2$ by $\|f_{\mathbf{w}+\mathbf{u}}(\mathbf{x} + \delta_{\mathbf{w}+\mathbf{u}}^{adv}(\mathbf{x})) - f_{\mathbf{w}}(\mathbf{x} + \delta_{\mathbf{w}}^{adv}(\mathbf{x}))\|_2$ (Farnia et al., 2018). For more details, see Appendix B.

## 6.2 Main Challenge

Based on Lemma 4, to provide an upper bound of robust margin loss is to solve the following problem:

**Problem 1.** *How to provide a bound of*

$$\|f_{\mathbf{w}+\mathbf{u}}(\mathbf{x} + \delta_{\mathbf{w}+\mathbf{u}}^{adv}(\mathbf{x})) - f_{\mathbf{w}}(\mathbf{x} + \delta_{\mathbf{w}}^{adv}(\mathbf{x}))\|_2? \tag{6}$$

We refer to the gap in Eq. (6) as robust weight perturbation. To the best of our knowledge, it remains unclear how to establish a bound for robust weight perturbation. In standard settings, when we perturb the weights from $\mathbf{w}$ to $\mathbf{w} + \mathbf{u}$, the input $\mathbf{x}$ remains the same. The change in function values is solely attributable to the change in weights. However, the situation becomes much more complex in adversarial settings. If we perturb the weights from $\mathbf{w}$ to $\mathbf{w} + \mathbf{u}$, the adversarial attacks also vary from $\delta_{\mathbf{w}}^{adv}(\mathbf{x})$ to $\delta_{\mathbf{w}+\mathbf{u}}^{adv}(\mathbf{x})$. The combined changes in input $\mathbf{x}$ and weights $\mathbf{w}$ may result in a substantial change in function values. The challenge of Problem 1 can be observed in previous studies.

Farnia et al. (2018) introduced additional assumptions to bound Eq. (6). For instance, for FGSM and PGM attacks, they assumed $|\nabla_{\mathbf{x}}\ell(f_{\mathbf{w}}(\mathbf{x}), y)| \geq \kappa$ for all $\mathbf{x}$ $\epsilon$-close to $\mathcal{X}$. This parameter $\kappa$ appears in the bound of Eq. (6) as well as in the final generalization bound. To the best of our knowledge, there has been no attempt at $\delta_{\mathbf{w}}^*(\mathbf{x})$. It is not because such research is unimportant (as mentioned in Sec. 3), but rather due to the challenge presented by Problem 1. In this case, it remains unclear what assumptions can be made to bound Eq. (6). The related work on Rademacher complexity analysis demonstrates the difficulty, as researchers have found it challenging to bound robust margin loss and have instead resorted to bounding robust loss against soled attack with additional assumptions. Further discussion on this topic can be found in Sec. 2.

---

[3]It is because $\|f_{\mathbf{w}+\mathbf{u}}(\mathbf{x}) - f_{\mathbf{w}}(\mathbf{x})\|_{\infty} \leq \|f_{\mathbf{w}+\mathbf{u}}(\mathbf{x}) - f_{\mathbf{w}}(\mathbf{x})\|_2$.

Our solution to this problem consists of two steps. Step 1: We recognize that a general and reasonable bound for Eq. (6) without additional assumptions may not exist. To address this, we establish a bound for a similar expression, namely the weight perturbation of margin operator, without requiring any additional assumptions. To develop this bound, we introduce a generalization framework called "Perturbation Bounds of Robustified Function", which can be further extended to analyze other neural network structures. Step 2: We modify Lemma 4 to incorporate the weight perturbation bound that we have introduced. By combining these two steps, we are able to address the challenges and provide a robust generalization bound.

## 6.3 Perturbation Bounds of Robustified Function

In this section, we consider functions $g_{\mathbf{w}}(\mathbf{x})$ parameterized by the weights of a neural network. We mainly consider scalar value functions $g_{\mathbf{w}}(\mathbf{x}) : \mathcal{X} \to \mathbb{R}$. For example, $g_{\mathbf{w}}(\mathbf{x})$ can be the $i^{th}$ output of a neural network $f_{\mathbf{w}}(\mathbf{x})[i]$, the margin operator $f_{\mathbf{w}}(\mathbf{x})[y] - \max_{j \neq y} f_{\mathbf{w}}(\mathbf{x})[j]$, or the robust margin operator.

**Definition 1** (Local Perturbation Bounds). Given $\mathbf{x} \in \mathcal{X}$, we say $g_{\mathbf{w}}(\mathbf{x})$ has a $(L_1, \cdots, L_d)$-local perturbation bound w.r.t. $\mathbf{w}$, if

$$|g_{\mathbf{w}}(\mathbf{x}) - g_{\mathbf{w}'}(\mathbf{x})| \leq \sum_{i=1}^{d} L_i \|W_i - W_i'\|, \tag{7}$$

where $L_i$ can be related to $\mathbf{w}$, $\mathbf{w}'$ and $\mathbf{x}$.

Eq. (7) controls the change of the output of functions $g_{\mathbf{w}}(\mathbf{x})$ given a slight perturbation on the weights of DNNs. The following Lemma is the key Lemma to estimate perturbation bounds of the robustified function, which is defined as $\inf_{\|\mathbf{x}-\mathbf{x}'\| \leq \epsilon} g_{\mathbf{w}}(\mathbf{x}')$. The reason why we require $g_{\mathbf{w}}(\mathbf{x})$ to be scalar functions is that we can define their corresponding robustified functions.

**Lemma 5 (Key Lemma).** *if $g_{\mathbf{w}}(\mathbf{x})$ has a $(A_1|\mathbf{x}|, \cdots, A_d|\mathbf{x}|)$-local perturbation bound, i.e.,*

$$|g_{\mathbf{w}}(\mathbf{x}) - g_{\mathbf{w}'}(\mathbf{x})| \leq \sum_{i=1}^{d} A_i|\mathbf{x}|\|W_i - W_i'\|,$$

*the robustified function $\inf_{\|\mathbf{x}-\mathbf{x}'\| \leq \epsilon} g_{\mathbf{w}}(\mathbf{x}')$ has a $(A_1(|\mathbf{x}| + \epsilon), \cdots, A_d(|\mathbf{x}| + \epsilon))$-local perturbation bound.*

Proof: Let $\mathbf{x}(\mathbf{w}) = \arg\inf_{\|\mathbf{x}-\mathbf{x}'\| \leq \epsilon} g_{\mathbf{w}}(\mathbf{x}')$, $\mathbf{x}(\mathbf{w}') = \arg\inf_{\|\mathbf{x}-\mathbf{x}'\| \leq \epsilon} g_{\mathbf{w}'}(\mathbf{x}')$, Then,

$$|\inf_{\|\mathbf{x}-\mathbf{x}'\| \leq \epsilon} g_{\mathbf{w}}(\mathbf{x}') - \inf_{\|\mathbf{x}-\mathbf{x}'\| \leq \epsilon} g_{\mathbf{w}'}(\mathbf{x}')| \leq \max\{|g_{\mathbf{w}}(\mathbf{x}(\mathbf{w})) - g_{\mathbf{w}'}(\mathbf{x}(\mathbf{w}))|, |g_{\mathbf{w}}(\mathbf{x}(\mathbf{w}')) - g_{\mathbf{w}'}(\mathbf{x}(\mathbf{w}'))|\}.$$

It is because $g_{\mathbf{w}}(\mathbf{x}(\mathbf{w})) - g_{\mathbf{w}'}(\mathbf{x}(\mathbf{w}')) \leq g_{\mathbf{w}}(\mathbf{x}(\mathbf{w}')) - g_{\mathbf{w}'}(\mathbf{x}(\mathbf{w}'))$ and $g_{\mathbf{w}'}(\mathbf{x}(\mathbf{w}')) - g_{\mathbf{w}}(\mathbf{x}(\mathbf{w})) \leq g_{\mathbf{w}'}(\mathbf{x}(\mathbf{w})) - g_{\mathbf{w}}(\mathbf{x}(\mathbf{w}))$. Therefore,

$$|\inf_{\|\mathbf{x}-\mathbf{x}'\| \leq \epsilon} g_{\mathbf{w}}(\mathbf{x}') - \inf_{\|\mathbf{x}-\mathbf{x}'\| \leq \epsilon} g_{\mathbf{w}'}(\mathbf{x}')| \leq \sum_{i=1}^{d} A_i|\mathbf{x}(\mathbf{w})|\|W_i - W_i'\| \leq \sum_{i=1}^{d} A_i(|\mathbf{x}| + \epsilon)\|W_i - W_i'\|.$$

$\square$

Lemma 5 shows that the local perturbation bound of the robustified function $\inf_{\|\mathbf{x}-\mathbf{x}'\| \leq \epsilon} g_{\mathbf{w}}(\mathbf{x}')$ can be estimated by the local perturbation bound of the function $g_{\mathbf{w}}(\mathbf{x})$, which is the key to provide robust generalization bounds.

## 6.4 Perturbation Bounds of Margin Operator

It should be noted that Lemma 5 is unable to provide a bound for Problem 1. In order to utilize Lemma 5, we shift our focus to the margin operator, which is a scalar function.

**Margin Operator.** Following the notation of (Bartlett et al., 2017), we define the margin operator of the true label $y$ given $\mathbf{x}$ and of a pair of two classes $(i, j)$ as

$$M(f_{\mathbf{w}}(\mathbf{x}), y) = f_{\mathbf{w}}(\mathbf{x})[y] - \max_{j \neq y} f_{\mathbf{w}}(\mathbf{x})[j], \ M(f_{\mathbf{w}}(\mathbf{x}), i, j) = f_{\mathbf{w}}(\mathbf{x})[i] - f_{\mathbf{w}}(\mathbf{x})[j].$$

**Robust Margin Operator.** Similarly, we define the robust margin operator of the true label $y$ and of a pair of two classes $(i,j)$ given $\mathbf{x}$ as

$$RM(f_\mathbf{w}(\mathbf{x}),y) = \inf_{\|\mathbf{x}-\mathbf{x}'\|\leq\epsilon}(f_\mathbf{w}(\mathbf{x}')[y] - \max_{j\neq y} f_\mathbf{w}(\mathbf{x}')[j]), \quad \text{and}$$

$$RM(f_\mathbf{w}(\mathbf{x}),i,j) = \inf_{\|\mathbf{x}-\mathbf{x}'\|\leq\epsilon}(f_\mathbf{w}(\mathbf{x}')[i] - f_\mathbf{w}(\mathbf{x}')[j]),$$

respectively. Based on Lemma 5, it is left to provide the form of $A_i$ for the margin operator.

**Lemma 6.** *Let $f_\mathbf{w}$ be a d-layer neural networks with Relu activation. The following local perturbation bounds hold.*

1. *Given $\mathbf{x}$ and $i,j$, the margin operator $M(f_\mathbf{w}(\mathbf{x}),i,j)$ has a $(A_1|\mathbf{x}|,\cdots,A_d|\mathbf{x}|)$-local perturbation bound w.r.t. $w$, where $A_i = 2e\prod_{l=1}^{d}\|W_l\|_2 / \|W_i\|_2$. And*

$$|M(f_{\mathbf{w}+\mathbf{u}}(\mathbf{x}),i,j) - M(f_\mathbf{w}(\mathbf{x}),i,j)| \leq 2eB\prod_{l=1}^{d}\|W_l\|_2 \sum_{i=1}^{d}\frac{\|U_i\|_2}{\|W_i\|_2}. \tag{8}$$

2. *Given $\mathbf{x}$ and $i,j$, the robust margin operator $RM(f_\mathbf{w}(\mathbf{x}),i,j)$ has a locally $(A_1(|\mathbf{x}|+\epsilon),\cdots,A_d(|\mathbf{x}|+\epsilon))$-local perturbation bound w.r.t. $w$. And*

$$|RM(f_{\mathbf{w}+\mathbf{u}}(\mathbf{x}),i,j) - RM(f_\mathbf{w}(\mathbf{x}),i,j)| \leq 2e(B+\epsilon)\prod_{l=1}^{d}\|W_l\|_2 \sum_{i=1}^{d}\frac{\|U_i\|_2}{\|W_i\|_2}. \tag{9}$$

The proof of Lemma 6.1 is adopted from Lemma 2 in (Neyshabur et al., 2017b), and the proof of Lemma 6.2 is a combination of Lemma 5 and Lemma 6.1. It is important to note that Eq. (9) provides a bound for a similar but different form of robust weight perturbation compared to Eq. (6), indicating that Problem 1 has not been fully resolved. However, we are fortunate that the subsequent lemma demonstrates that Eq. (9) is sufficient to yield the final robust generalization bound.

**Lemma 7.** *Let $f_\mathbf{w}(\mathbf{x}) : \mathcal{X} \to \mathbb{R}^k$ be any predictor with parameters $\mathbf{w}$, and $P$ be any distribution on the parameters that is independent of the training data. Then, for any $\gamma, \delta > 0$, with probability $\geq 1 - \delta$ over the training set of size $m$, for any $\mathbf{w}$, and any random perturbation $\mathbf{u}$ s.t.*

1. $\mathbb{P}_\mathbf{u}[\max_{i,j\in[k],\mathbf{x}\in\mathcal{X}}|M(f_{\mathbf{w}+\mathbf{u}}(\mathbf{x}),i,j) - M(f_\mathbf{w}(\mathbf{x}),i,j)| < \frac{\gamma}{2}] \geq \frac{1}{2}$, *we have:*

$$L_0(f_\mathbf{w}) \leq \widehat{L}_\gamma(f_\mathbf{w}) + 4\sqrt{\frac{KL(\mathbf{w}+\mathbf{u}\|P) + \ln\frac{6m}{\delta}}{m-1}}.$$

2. $\mathbb{P}_\mathbf{u}[\max_{i,j\in[k],\mathbf{x}\in\mathcal{X}}|RM(f_{\mathbf{w}+\mathbf{u}}(\mathbf{x}),i,j) - RM(f_\mathbf{w}(\mathbf{x}),i,j)| < \frac{\gamma}{2}] \geq \frac{1}{2}$, *we have:*

$$R_0(f_\mathbf{w}) \leq \hat{R}_\gamma(f_\mathbf{w}) + 4\sqrt{\frac{KL(\mathbf{w}+\mathbf{u}\|P) + \ln\frac{6m}{\delta}}{m-1}}.$$

**Remark:** Lemma 7 shows that we can replace the robust weight perturbation (Eq. (6)) by the weight perturbation of the robust margin operator. The proof is deferred to the Appendix.

Now that we have established the complete framework of the *perturbation bound of robustified function* to derive the robust generalization bound, we are ready to prove Theorem 1. By following the proof of (Neyshabur et al., 2017b), we can replicate the standard generalization bound by combining Lemma 6.1 and 7.1. Similarly, we can obtain the robust generalization bound by combining Lemma 6.2 and 7.2. The flowchart illustrating this process is presented in Figure 2. Additionally, Lemma 5 serves as a crucial link between the robust margin operator and the margin operator, thus establishing the connection between the robust generalization bound and the standard generalization bound.

# 7 Extension of the Main Result

The provided framework allows us to extend the result to 1) general non-$\ell_p$ adversarial attacks and 2) other neural network structures.

**Extension to Non-$\ell_p$ Adversarial Attacks.** Even though most of the adversarial robustness studies focused on norm-bounded attacks, real-world attacks are not restricted in the $\ell_p$-ball. We consider the following general adversarial attack problem:

$$\max_{\mathbf{x}' \in C(\mathbf{x})} \ell(f_{\mathbf{w}}(\mathbf{x}'), y),$$

where $C(\mathbf{x})$ can be any reasonable constraint given the original example $\mathbf{x}$. Assume that $\max_{x \in S} \max_{\mathbf{x}' \in C(\mathbf{x})} |\mathbf{x}'| = D$. In words, the norm of the adversarial examples is bounded by $D$.

**Theorem 8** (Robust Generalization Bound for non-$\ell_p$ attack.)**.** *For any $D, d, h$, let $f_{\mathbf{w}} : \mathcal{X} \to \mathbb{R}^k$ be a $d$-layer feedforward network with ReLU activations. Then, for any $\delta, \gamma > 0$, with probability $\geq 1 - \delta$ over a training set of size $m$, for any $\mathbf{w}$, we have:*

$$R_0^{nl}(f_{\mathbf{w}}) - \hat{R}_\gamma^{nl}(f_{\mathbf{w}}) \leq \mathcal{O}\left(\sqrt{\frac{D^2 d^2 h \ln(dh)\Phi(f_{\mathbf{w}}) + \ln \frac{dm}{\delta}}{\gamma^2 m}}\right),$$

*where $\Phi(f_{\mathbf{w}}) = \Pi_{i=1}^d \|W_i\|_2^2 \sum_{i=1}^d \frac{\|W_i\|_F^2}{\|W_i\|_2^2}$ and nl stands for non-$\ell_p$ adversarial attacks.*

The proof is based on a slight modification of Lemma 5.

**Extension to Other Neural Networks Structure.** The framework we have established enables us to extend the PAC-Bayesian generalization bound from standard settings to robust settings, provided that the standard generalization bound is also obtained using this framework. Importantly, this extension is independent of the structure of the neural networks.

**ResNet.** Consider a neural network: $f_{\mathbf{w}}^1(\mathbf{x}) = W_1 \mathbf{x}$ and $f_{\mathbf{w}}^i(\mathbf{x}) = W_i \phi(f_{\mathbf{w}}^{i-1}(\mathbf{x})) + f_{\mathbf{w}}^{i-1}(\mathbf{x})$. ResNet in practice could be complicated. We use this structure for illustration.

**Theorem 9** (Robust Generalization Bound for ResNet)**.** *For any $D, d, h$, let $f_{\mathbf{w}} : \mathcal{X} \to \mathbb{R}^k$ be a $d$-layer ResNet with ReLU activations. Then, for any $\delta, \gamma > 0$, with probability $\geq 1 - \delta$ over a training set of size $m$, for any $\mathbf{w}$, we have:*

$$R_0(f_{RN}) - \hat{R}_\gamma(f_{RN}) \leq \mathcal{O}\left(\sqrt{\frac{(B+\epsilon)^2 d^2 h \ln(dh)\Phi(f_{RN}) + \ln \frac{dm}{\delta}}{\gamma^2 m}}\right),$$

*where $\Phi(f_{RN}) = \Pi_{i=1}^d (\|W_i\|_2 + 1)^2 \sum_{i=1}^d \frac{\|W_i\|_F^2}{(\|W_i\|_2 + 1)^2}$.*

# 8 Conclusion

**Limitation.** The primary limitation lies in the fact that norm-based bounds tend to be excessively large in practical scenarios. As illustrated in Table 1, the bounds for VGG networks surpass $10^9$ in the experiments on CIFAR-10 dataset. The challenge at hand is how to achieve smaller norm-based bounds in practical contexts, not only in adversarial settings but also in standard settings. This remains an open problem.

In this paper, we introduce a PAC-Bayesian spectrally-normalized robust generalization bound. The proof is constructed based on the framework of the perturbation bound of the robustified function. This established framework enables us to extend the generalization bound from standard settings to robust settings, as well as to generalize the results to encompass various adversarial attacks and DNN architectures. The simplicity of this framework makes it a valuable tool for analyzing robust generalization in machine learning.

## Acknowledgement

We would like to thank all the anonymous reviewers for their comments and suggestions. This paper is supported in part by the National Key Research and Development Project under grant 2022YFA1003900; Hetao Shenzhen-Hong Kong Science and Technology Innovation Cooperation Zone Project (No.HZQSWS-KCCYB-2022046); Guangdong Key Lab on the Mathematical Foundation of Artificial Intelligence, Department of Science and Techonology of Guangdong Province. The work of Zhi-Quan Luo is supported by NSFC-617310018 and the Guangdong Provincial Key Laboratory of Big Data Computing.

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

# A Proof of Theorems

The proof of the key lemma (Lemma 5), which establishes a connection between the margin operator and the robust margin operator, is presented in the main content.

We still need to demonstrate that the properties in PAC-Bayes analysis hold for both the margin operator and the robust margin operator. The following proofs are adapted from the work of (Neyshabur et al., 2017b), with the steps being kept independent of the (robust) margin operator. We will begin by finishing the proofs of Lemma 6 and Lemma 7. Afterward, we will proceed to complete the proof of Theorem 1, which is our primary result.

## A.1 Proof of Lemma 6

Proof of Lemma 6.1:

For any $i \in [k]$,

$$|f_{\mathbf{w}+\mathbf{u}}(\mathbf{x})[i] - f_{\mathbf{w}}(\mathbf{x})[i]| \leq \|f_{\mathbf{w}+\mathbf{u}}(\mathbf{x}) - f_{\mathbf{w}}(\mathbf{x})\|_2.$$

For any $i, j \in [k]$,

$$|M(f_{\mathbf{w}+\mathbf{u}}(\mathbf{x}), i, j) - M(f_{\mathbf{w}}(\mathbf{x}), i, j)| \leq 2|f_{\mathbf{w}+\mathbf{u}}(\mathbf{x})[i] - f_{\mathbf{w}}(\mathbf{x})[i]| \leq 2\|f_{\mathbf{w}+\mathbf{u}}(\mathbf{x}) - f_{\mathbf{w}}(\mathbf{x})\|_2.$$

Therefore, it is left to bound $\|f_{\mathbf{w}+\mathbf{u}}(\mathbf{x}) - f_{\mathbf{w}}(\mathbf{x})\|$. It is provided in (Neyshabur et al., 2017b), we provide the proof here for reference. Let $\Delta_i = \left|f^i_{\mathbf{w}+\mathbf{u}}(\mathbf{x}) - f^i_{\mathbf{w}}(\mathbf{x})\right|_2$. We will prove using induction that for any $i \geq 0$:

$$\Delta_i \leq \left(1 + \frac{1}{d}\right)^i \left(\prod_{j=1}^i \|W_j\|_2\right) |\mathbf{x}|_2 \sum_{j=1}^i \frac{\|U_j\|_2}{\|W_j\|_2}.$$

The above inequality together with $\left(1 + \frac{1}{d}\right)^d \leq e$ proves the lemma statement. The induction base clearly holds since $\Delta_0 = |\mathbf{x} - \mathbf{x}|_2 = 0$. For any $i \geq 1$, we have the following:

$$
\begin{aligned}
\Delta_{i+1} &= \left|(W_{i+1} + U_{i+1})\, \phi_i(f^i_{\mathbf{w}+\mathbf{u}}(\mathbf{x})) - W_{i+1}\phi_i(f^i_{\mathbf{w}}(\mathbf{x}))\right|_2 \\
&= \left|(W_{i+1} + U_{i+1})\left(\phi_i(f^i_{\mathbf{w}+\mathbf{u}}(\mathbf{x})) - \phi_i(f^i_{\mathbf{w}}(\mathbf{x}))\right) + U_{i+1}\phi_i(f^i_{\mathbf{w}}(\mathbf{x}))\right|_2 \\
&\leq (\|W_{i+1}\|_2 + \|U_{i+1}\|_2)\left|\phi_i(f^i_{\mathbf{w}+\mathbf{u}}(\mathbf{x})) - \phi_i(f^i_{\mathbf{w}}(\mathbf{x}))\right|_2 + \|U_{i+1}\|_2\left|\phi_i(f^i_{\mathbf{w}}(\mathbf{x}))\right|_2 \\
&\leq (\|W_{i+1}\|_2 + \|U_{i+1}\|_2)\left|f^i_{\mathbf{w}+\mathbf{u}}(\mathbf{x}) - f^i_{\mathbf{w}}(\mathbf{x})\right|_2 + \|U_{i+1}\|_2\left|f^i_{\mathbf{w}}(\mathbf{x})\right|_2 \\
&= \Delta_i\left(\|W_{i+1}\|_2 + \|U_{i+1}\|_2\right) + \|U_{i+1}\|_2\left|f^i_{\mathbf{w}}(\mathbf{x})\right|_2,
\end{aligned}
$$

where the last inequality is by the Lipschitz property of the activation function and using $\phi(0) = 0$. The $\ell_2$ norm of outputs of layer $i$ is bounded by $|\mathbf{x}|_2 \Pi_{j=1}^i \|W_j\|_2$ and by the lemma assumption we have $\|U_{i+1}\|_2 \leq \frac{1}{d}\|W_{i+1}\|_2$. Therefore, using the induction step, we get the following bound:

$$
\begin{aligned}
\Delta_{i+1} &\leq \Delta_i\left(1 + \frac{1}{d}\right)\|W_{i+1}\|_2 + \|U_{i+1}\|_2 |\mathbf{x}|_2 \prod_{j=1}^i \|W_j\|_2 \\
&\leq \left(1 + \frac{1}{d}\right)^{i+1}\left(\prod_{j=1}^{i+1}\|W_j\|_2\right)|\mathbf{x}|_2 \sum_{j=1}^i \frac{\|U_j\|_2}{\|W_j\|_2} + \frac{\|U_{i+1}\|_2}{\|W_{i+1}\|_2}|\mathbf{x}|_2 \prod_{j=1}^{i+1}\|W_i\|_2 \\
&\leq \left(1 + \frac{1}{d}\right)^{i+1}\left(\prod_{j=1}^{i+1}\|W_j\|_2\right)|\mathbf{x}|_2 \sum_{j=1}^{i+1} \frac{\|U_j\|_2}{\|W_j\|_2}.
\end{aligned}
$$

Then we complete the proof of Lemma 6.1. By combining Lemma 6.1 and Lemma 5, we directly obtain Lemma 6.2.

$\square$

## A.2 Proof of Lemma 7

The proof of Lemma 7.1 and 7.2 is similar. We provide the proof of Lemma 7.2 below. The proof of Lemma 7.1 follows the proof of Lemma 7.2 by replacing the robust margin operator by the margin operator.

Let $\mathbf{w}' = \mathbf{w} + \mathbf{u}$. Let $\mathcal{S}_{\mathbf{w}}$ be the set of perturbations with the following property:

$$\mathcal{S}_{\mathbf{w}} \subseteq \left\{ \mathbf{w}' \, \middle| \, \max_{i,j \in [k], \mathbf{x} \in \mathcal{X}} |RM(f_{\mathbf{w}'}(\mathbf{x}), i, j) - RM(f_{\mathbf{w}}(\mathbf{x}), i, j)| < \frac{\gamma}{2} \right\}.$$

Let $q$ be the probability density function over the parameters $\mathbf{w}'$. We construct a new distribution $\tilde{Q}$ over predictors $f_{\tilde{\mathbf{w}}}$ where $\tilde{\mathbf{w}}$ is restricted to $\mathcal{S}_{\mathbf{w}}$ with the probability density function:

$$\tilde{q}(\tilde{\mathbf{w}}) = \frac{1}{Z} \begin{cases} q(\tilde{\mathbf{w}}) & \tilde{\mathbf{w}} \in \mathcal{S}_{\mathbf{w}} \\ 0 & \text{otherwise.} \end{cases}$$

Here $Z$ is a normalizing constant and by the lemma assumption $Z = \mathbb{P}\left[\mathbf{w}' \in \mathcal{S}_{\mathbf{w}}\right] \geq \frac{1}{2}$. By the definition of $\tilde{Q}$, we have:

$$\max_{i,j \in [k], \mathbf{x} \in \mathcal{X}} |RM(f_{\tilde{\mathbf{w}}}(\mathbf{x}), i, j) - RM(f_{\mathbf{w}}(\mathbf{x}), i, j)| < \frac{\gamma}{2}.$$

Since the above bound holds for any $\mathbf{x}$ in the domain $\mathcal{X}$, we can get the following a.s.:

$$R_0(f_{\mathbf{w}}) \leq R_{\frac{\gamma}{2}}(f_{\tilde{\mathbf{w}}})$$
$$\hat{R}_{\frac{\gamma}{2}}(f_{\tilde{\mathbf{w}}}) \leq \hat{R}_{\gamma}(f_{\mathbf{w}})$$

Now using the above inequalities together with the equation (5), with probability $1 - \delta$ over the training set we have:

$$R_0(f_{\mathbf{w}}) \leq \mathbb{E}_{\tilde{\mathbf{w}}}\left[R_{\frac{\gamma}{2}}(f_{\tilde{\mathbf{w}}})\right]$$

$$\leq \mathbb{E}_{\tilde{\mathbf{w}}}\left[\hat{R}_{\frac{\gamma}{2}}(f_{\tilde{\mathbf{w}}})\right] + 2\sqrt{\frac{2(KL\left(\tilde{\mathbf{w}}\|P\right) + \ln\frac{2m}{\delta})}{m - 1}}$$

$$\leq \hat{R}_{\gamma}(f_{\mathbf{w}}) + 2\sqrt{\frac{2(KL\left(\tilde{\mathbf{w}}\|P\right) + \ln\frac{2m}{\delta})}{m - 1}}$$

$$\leq \hat{R}_{\gamma}(f_{\mathbf{w}}) + 4\sqrt{\frac{KL\left(\mathbf{w}'\|P\right) + \ln\frac{6m}{\delta}}{m - 1}},$$

The last inequality follows from the following calculation.

Let $\mathcal{S}_{\mathbf{w}}^c$ denote the complement set of $\mathcal{S}_{\mathbf{w}}$ and $\tilde{q}^c$ denote the density function $q$ restricted to $\mathcal{S}_{\mathbf{w}}^c$ and normalized. Then,

$$KL(q\|p) = ZKL(\tilde{q}\|p) + (1 - Z)KL(\tilde{q}^c\|p) - H(Z),$$

where $H(Z) = -Z\ln Z - (1 - Z)\ln(1 - Z) \leq 1$ is the binary entropy function. Since KL is always positive, we get,

$$KL(\tilde{q}\|p) = \frac{1}{Z}\left[KL(q\|p) + H(Z)) - (1 - Z)KL(\tilde{q}^c\|p)\right] \leq 2(KL(q\|p) + 1).$$

## A.3 Proof of Theorem 1

Given the local perturbation bound of the robust margin operator and Lemma 5, the proof of Theorem 1 follows the procedure of the proof of Theorem 2.

Let $\beta = \left(\prod_{i=1}^{d} \|W_i\|_2\right)^{1/d}$ and consider a network with the normalized weights $\widetilde{W}_i = \frac{\beta}{\|W_i\|_2} W_i$. Due to the homogeneity of the ReLU, we have that for feedforward networks with ReLU activations

$f_{\widetilde{\mathbf{w}}} = f_{\mathbf{w}}$, and so the (empirical and expected) loss (including margin loss) is the same for $\mathbf{w}$ and $\widetilde{\mathbf{w}}$. We can also verify that $\left(\prod_{i=1}^d \|W_i\|_2\right) = \left(\prod_{i=1}^d \left\|\widetilde{W_i}\right\|_2\right)$ and $\frac{\|W_i\|_F}{\|W_i\|_2} = \frac{\|\tilde{W}_i\|_F}{\|\tilde{W}_i\|_2}$, and so the excess error in the Theorem statement is also invariant to this transformation. It is therefore sufficient to prove the Theorem only for the normalized weights $\tilde{\mathbf{w}}$, and hence we assume w.l.o.g. that the spectral norm is equal across layers, i.e. for any layer $i$, $\|W_i\|_2 = \beta$.

Choose the distribution of the prior $P$ to be $\mathcal{N}(0, \sigma^2 I)$, and consider the random perturbation $\mathbf{u} \sim \mathcal{N}(0, \sigma^2 I)$, with the same $\sigma$, which we will set later according to $\beta$. More precisely, since the prior cannot depend on the learned predictor $\mathbf{w}$ or its norm, we will set $\sigma$ based on an approximation $\tilde{\beta}$. For each value of $\tilde{\beta}$ on a pre-determined grid, we will compute the PAC-Bayes bound, establishing the generalization guarantee for all $\mathbf{w}$ for which $|\beta - \tilde{\beta}| \le \frac{1}{d}\beta$, and ensuring that each relevant value of $\beta$ is covered by some $\tilde{\beta}$ on the grid. We will then take a union bound over all $\tilde{\beta}$ on the grid. For now, we will consider a fixed $\tilde{\beta}$ and the $\mathbf{w}$ for which $|\beta - \tilde{\beta}| \le \frac{1}{d}\beta$, and hence $\frac{1}{e}\beta^{d-1} \le \tilde{\beta}^{d-1} \le e\beta^{d-1}$.

Since $\mathbf{u} \sim \mathcal{N}(0, \sigma^2 I)$, we get the following bound for the spectral norm of $U_i$ (Tropp, 2012):
$$\mathbb{P}_{U_i \sim N(0,\sigma^2 I)}\left[\|U_i\|_2 > t\right] \le 2he^{-t^2/2h\sigma^2}.$$
Taking a union bond over the layers, we get that, with probability $\ge \frac{1}{2}$, the spectral norm of the perturbation $U_i$ in each layer is bounded by $\sigma\sqrt{2h\ln(4dh)}$. Plugging this spectral norm bound into the Lipschitz of robust margin operator we have that with probability at least $\frac{1}{2}$,

$$\max_{i,j\in[k],\mathbf{x}\in\mathcal{X}} |RM(f_{\mathbf{w}'}(\mathbf{x}), i, j) - RM(f_{\mathbf{w}}(\mathbf{x}), i, j)| \tag{10}$$

$$\le 2e(B+\epsilon)\beta^d \sum_i \frac{\|U_i\|_2}{\beta}$$

$$= e(B+\epsilon)\beta^{d-1} \sum_i \|U_i\|_2 \le e^2 d(B+\epsilon)\tilde{\beta}^{d-1}\sigma\sqrt{2h\ln(4dh)} \le \frac{\gamma}{2}, \tag{11}$$

where we choose $\sigma = \frac{\gamma}{42d(B+\epsilon)\tilde{\beta}^{d-1}\sqrt{h\ln(4hd)}}$ to get the last inequality, the first inequality is Lemma 6.2. The second inequality is the tail bound above. Hence, the perturbation $\mathbf{u}$ with the above value of $\sigma$ satisfies the assumptions of the Lemma 4.

We now calculate the KL-term in Lemma 4 with the chosen distributions for $P$ and $\mathbf{u}$, for the above value of $\sigma$.
$$KL(\mathbf{w}+\mathbf{u}\|P)$$
$$\le \frac{|\mathbf{w}|^2}{2\sigma^2} = \frac{42^2 d^2(B+\epsilon)^2\tilde{\beta}^{2d-2}h\ln(4hd)}{2\gamma^2} \sum_{i=1}^d \|W_i\|_F^2$$
$$\le \mathcal{O}\left((B+\epsilon)^2 d^2 h\ln(dh)\frac{\beta^{2d}}{\gamma^2}\sum_{i=1}^d \frac{\|W_i\|_F^2}{\beta^2}\right)$$
$$\le \mathcal{O}\left((B+\epsilon)^2 d^2 h\ln(dh)\frac{\Pi_{i=1}^d \|W_i\|_2^2}{\gamma^2}\sum_{i=1}^d \frac{\|W_i\|_F^2}{\|W_i\|_2^2}\right).$$

Hence, for any $\tilde{\beta}$, with probability $\ge 1-\delta$ and for all $\mathbf{w}$ such that, $|\beta - \tilde{\beta}| \le \frac{1}{d}\beta$, we have:

$$R_0(f_{\mathbf{w}}) \le \hat{R}_\gamma(f_{\mathbf{w}}) + \mathcal{O}\left(\sqrt{\frac{(B+\epsilon)^2 d^2 h\ln(dh)\Pi_{i=1}^d \|W_i\|_2^2 \sum_{i=1}^d \frac{\|W_i\|_F^2}{\|W_i\|_2^2} + \ln\frac{m}{\delta}}{\gamma^2 m}}\right). \tag{12}$$

For other $\ell_p$ attacks, the results are directly obtained by Lemma 4 of (Xiao et al., 2022a).

### A.4 Proof of Theorem 8

It is based on a slight modification of the key lemma. if $g_{\mathbf{w}}(\mathbf{x})$ has a $(A_1|\mathbf{x}|, \cdots, A_d|\mathbf{x}|)$-local perturbation bound, *i.e.,*

$$|g_{\mathbf{w}}(\mathbf{x}) - g_{\mathbf{w}'}(\mathbf{x})| \le \sum_{i=1}^d A_i|\mathbf{x}|\|W_i - W_i'\|,$$

the robustified function $\inf_{\mathbf{x}' \in C(\mathbf{x})} g_{\mathbf{w}}(\mathbf{x}')$ has a $(A_1 D, \cdots, A_d D)$-local perturbation bound.

Proof: Let
$$\mathbf{x}(\mathbf{w}) = \arg \inf_{\mathbf{x}' \in C(\mathbf{x})} g_{\mathbf{w}}(\mathbf{x}'),$$
$$\mathbf{x}(\mathbf{w}') = \arg \inf_{\mathbf{x}' \in C(\mathbf{x})} g_{\mathbf{w}'}(\mathbf{x}'),$$

Then,
$$| \inf_{\|\mathbf{x}-\mathbf{x}'\| \le \epsilon} g_{\mathbf{w}}(\mathbf{x}') - \inf_{\|\mathbf{x}-\mathbf{x}'\| \le \epsilon} g_{\mathbf{w}'}(\mathbf{x}')| \le$$
$$\max\{|g_{\mathbf{w}}(\mathbf{x}(\mathbf{w})) - g_{\mathbf{w}'}(\mathbf{x}(\mathbf{w}))|, |g_{\mathbf{w}}(\mathbf{x}(\mathbf{w}')) - g_{\mathbf{w}'}(\mathbf{x}(\mathbf{w}'))|\}.$$

It is because $g_{\mathbf{w}}(\mathbf{x}(\mathbf{w})) - g_{\mathbf{w}'}(\mathbf{x}(\mathbf{w}')) \le g_{\mathbf{w}}(\mathbf{x}(\mathbf{w}')) - g_{\mathbf{w}'}(\mathbf{x}(\mathbf{w}'))$ and $g_{\mathbf{w}'}(\mathbf{x}(\mathbf{w}')) - g_{\mathbf{w}}(\mathbf{x}(\mathbf{w})) \le g_{\mathbf{w}'}(\mathbf{x}(\mathbf{w})) - g_{\mathbf{w}}(\mathbf{x}(\mathbf{w}))$. Therefore,

$$| \inf_{\|\mathbf{x}-\mathbf{x}'\| \le \epsilon} g_{\mathbf{w}}(\mathbf{x}') - \inf_{\|\mathbf{x}-\mathbf{x}'\| \le \epsilon} g_{\mathbf{w}'}(\mathbf{x}')|$$
$$\le \sum_{i=1}^{d} A_i |\mathbf{x}(\mathbf{w})| \|W_i - W_i'\|$$
$$\le \sum_{i=1}^{d} A_i D \|W_i - W_i'\|.$$

Therefore, combining the local perturbation bound and Lemma 7.2, we complete the proof. $\square$

### A.5  Proof of Theorem 9

As shown in the proof of Lemma 6, it is left to bound $\|f_{\mathbf{w}+\mathbf{u}}(\mathbf{x}) - f_{\mathbf{w}}(\mathbf{x})\|$. Let $\Delta_i = |f_{\mathbf{w}+\mathbf{u}}^i(\mathbf{x}) - f_{\mathbf{w}}^i(\mathbf{x})|_2$. We will prove using induction that for any $i \ge 0$:

$$\Delta_i \le \left(1 + \frac{1}{d}\right)^i \left(\prod_{j=1}^{i}(\|W_j\|_2 + 1)\right) |\mathbf{x}|_2 \sum_{j=1}^{i} \frac{\|U_j\|_2}{(\|W_j\|_2 + 1)}.$$

The above inequality together with $\left(1 + \frac{1}{d}\right)^d \le e$ proves the lemma statement. The induction base clearly holds since $\Delta_0 = |\mathbf{x} - \mathbf{x}|_2 = 0$. For any $i \ge 1$, we have the following:

$$\Delta_{i+1} = \left|(W_{i+1} + U_{i+1}) \phi_i(f_{\mathbf{w}+\mathbf{u}}^i(\mathbf{x})) - W_{i+1}\phi_i(f_{\mathbf{w}}^i(\mathbf{x})) + (f_{\mathbf{w}+\mathbf{u}}^i(\mathbf{x}) - f_{\mathbf{w}}^i(\mathbf{x}))\right|_2$$
$$= \left|(W_{i+1} + U_{i+1}) \left(\phi_i(f_{\mathbf{w}+\mathbf{u}}^i(\mathbf{x})) - \phi_i(f_{\mathbf{w}}^i(\mathbf{x}))\right) + U_{i+1}\phi_i(f_{\mathbf{w}}^i(\mathbf{x})) + (f_{\mathbf{w}+\mathbf{u}}^i(\mathbf{x}) - f_{\mathbf{w}}^i(\mathbf{x}))\right|_2$$
$$\le (\|W_{i+1}\|_2 + \|U_{i+1}\|_2) \left|\phi_i(f_{\mathbf{w}+\mathbf{u}}^i(\mathbf{x})) - \phi_i(f_{\mathbf{w}}^i(\mathbf{x}))\right|_2 + \|U_{i+1}\|_2 \left|\phi_i(f_{\mathbf{w}}^i(\mathbf{x}))\right|_2 + \Delta_i$$
$$\le (\|W_{i+1}\|_2 + \|U_{i+1}\|_2) \left|f_{\mathbf{w}+\mathbf{u}}^i(\mathbf{x}) - f_{\mathbf{w}}^i(\mathbf{x})\right|_2 + \|U_{i+1}\|_2 \left|f_{\mathbf{w}}^i(\mathbf{x})\right|_2 + \Delta_i$$
$$= \Delta_i (\|W_{i+1}\|_2 + \|U_{i+1}\|_2 + 1) + \|U_{i+1}\|_2 \left|f_{\mathbf{w}}^i(\mathbf{x})\right|_2,$$

where the last inequality is by the Lipschitz property of the activation function and using $\phi(0) = 0$. The $\ell_2$ norm of outputs of layer $i$ is bounded by $|\mathbf{x}|_2 \Pi_{j=1}^{i}(\|W_j\|_2 + 1)$ and by the lemma assumption we have $\|U_{i+1}\|_2 \le \frac{1}{d} \|W_{i+1}\|_2$. Therefore, using the induction step, we get the following bound:

$$\Delta_{i+1} \le \Delta_i \left(1 + \frac{1}{d}\right) (\|W_{i+1}\|_2 + 1) + \|U_{i+1}\|_2 |\mathbf{x}|_2 \prod_{j=1}^{i}(\|W_j\|_2 + 1)$$
$$\le \left(1 + \frac{1}{d}\right)^{i+1} \left(\prod_{j=1}^{i+1}(\|W_j\|_2 + 1)\right) |\mathbf{x}|_2 \sum_{j=1}^{i} \frac{\|U_j\|_2}{(\|W_j\|_2 + 1)} + \frac{\|U_{i+1}\|_2}{(\|W_{i+1}\|_2 + 1)} |\mathbf{x}|_2 \prod_{j=1}^{i+1}(\|W_i\|_2 + 1)$$
$$\le \left(1 + \frac{1}{d}\right)^{i+1} \left(\prod_{j=1}^{i+1}(\|W_j\|_2 + 1)\right) |\mathbf{x}|_2 \sum_{j=1}^{i+1} \frac{\|U_j\|_2}{(\|W_j\|_2 + 1)}.$$

Therefore, the margin operator of ResNet is locally $(A_1 |\mathbf{x}|, \cdots, A_d |\mathbf{x}|)$-Lipschitz w.r.t. $w$, where

$$A_i = 2e \prod_{l=1}^{d}(\|W_l\|_2 + 1)/(\|W_i\|_2 + 1).$$

For any $\delta, \gamma > 0$, with probability $\geq 1 - \delta$ over a training set of size $m$, for any $\mathbf{w}$, we have:

$$L_0(f_{\mathrm{RN}}) - \hat{L}_\gamma(f_{\mathrm{RN}})$$
$$\leq \mathcal{O}\left(\sqrt{\frac{B^2 d^2 h \ln(dh) \Phi(f_{\mathrm{RN}}) + \ln\frac{dm}{\delta}}{\gamma^2 m}}\right);$$

By a combination of Lemma 5 and Lemma 7, for any $\delta, \gamma > 0$, with probability $\geq 1 - \delta$ over a training set of size $m$, for any $\mathbf{w}$, we have:

$$R_0(f_{\mathrm{RN}}) - \hat{R}_\gamma(f_{\mathrm{RN}})$$
$$\leq \mathcal{O}\left(\sqrt{\frac{(B+\epsilon)^2 d^2 h \ln(dh) \Phi(f_{\mathrm{RN}}) + \ln\frac{dm}{\delta}}{\gamma^2 m}}\right),$$

where $\Phi(f_{\mathrm{RN}}) = \Pi_{i=1}^d (\|W_i\|_2 + 1)^2 \sum_{i=1}^d \frac{\|W_i\|_F^2}{(\|W_i\|_2 + 1)^2}$. $\qquad \square$

# B  PAC-Bayesian Framework for Robust Generalization

PAC-Bayes analysis (McAllester, 1999) is a framework to provide generalization guarantees for randomized predictors drawn from a learned distribution $Q$ (as opposed to a single predictor) that depends on the training data set. The expected generalization gap over the posterior distribution $Q$ can be bounded in terms of the Kullback-Leibler divergence between the prior distribution $P$ and the posterior distribution $Q$, $KL(P\|Q)$.

A direct corollary of Eq. (5) is that, the expected robust error of $f_{\mathbf{w}+\mathbf{u}}$ can be bounded as follows

$$\mathbb{E}_{\mathbf{u}}[R_0^{adv}(f_{\mathbf{w}+\mathbf{u}})]$$
$$\leq \mathbb{E}_{\mathbf{u}}[\hat{R}_0^{adv}(f_{\mathbf{w}+\mathbf{u}})] + 2\sqrt{\frac{2\left(KL\left(\mathbf{w}+\mathbf{u}\|P\right) + \ln\frac{2m}{\delta}\right)}{m-1}}. \tag{13}$$

By a slight modification of Lemma 4, the following lemma given in the work of (Farnia et al., 2018) shows how to obtain an robust generalization bound.

**Lemma 10** (Farnia et al. (2018)). *Let $f_{\mathbf{w}}(\mathbf{x}) : \mathcal{X} \to \mathbb{R}^k$ be any predictor (not necessarily a neural network) with parameters $\mathbf{w}$, and $P$ be any distribution on the parameters that is independent of the training data. Then, for any $\gamma, \delta > 0$, with probability $\geq 1 - \delta$ over the training set of size $m$, for any $\mathbf{w}$, and any random perturbation $\mathbf{u}$ s.t. $\mathbb{P}_{\mathbf{u}}[\max_{\mathbf{x} \in \mathcal{X}} \left|f_{\mathbf{w}+\mathbf{u}}(\mathbf{x} + \delta_{\mathbf{w}+\mathbf{u}}^{adv}(\mathbf{x})) - f_{\mathbf{w}}(\mathbf{x} + \delta_{\mathbf{w}}^{adv}(\mathbf{x}))\right|_\infty < \frac{\gamma}{4}] \geq \frac{1}{2}$, we have:*

$$R_0^{adv}(f_{\mathbf{w}}) \leq \hat{R}_\gamma^{adv}(f_{\mathbf{w}}) + 4\sqrt{\frac{KL\left(\mathbf{w}+\mathbf{u}\|P\right) + \ln\frac{6m}{\delta}}{m-1}}.$$

Table 1: Comparison of the empirical results of the standard generalization bound and robust generalization in the experiment of training MNIST, CIFAR-10 and CIFAR-100 on VGG networks.

|  | MNIST | CIFAR-10 | CIFAR-100 |
|---|---|---|---|
| Standard Generalization Gap | 1.13% | 9.21% | 23.61% |
| Bound in Theorem 2 (Neyshabur et al., 2017b) | $1.33 \times 10^4$ | $1.34 \times 10^9$ | $3.41 \times 10^{11}$ |
| Robust Generalization Gap | 9.67% | 51.41% | 78.82% |
| Bound in Theorem 3 (Farnia et al., 2018) | NA | NA | NA |
| Bound in Theorem 1 (Ours) | $3.23 \times 10^4$ | $5.97 \times 10^{10}$ | $1.66 \times 10^{13}$ |

# C  Empirical Study of the Generalization Bounds

The spectral complexity $\Phi(f_{\mathbf{w}})$ induced by adversarial training is significantly larger. We conducted experiments training MNIST, CIFAR-10, and CIFAR-100 datasets using VGG-19 networks, following

the training parameters described in (Neyshabur et al., 2017a).[4] The results are presented in Table 1. It is evident that adversarial training can induce a larger spectral complexity, resulting in a larger generalization bound.[5] We refer the readers to our previous work (Xiao et al., 2022a) for more experiments results about norm-based complexity of adversarially-trained models. These experiments align with the findings presented by (Bartlett et al., 2017), indicating: 1) spectral complexity scales with the difficulty of the learning task, and 2) the generalization bound is sensitive to this complexity.

---

[4]The settings of standard training follows the experiments in `https://github.com/bneyshabur/generalization-bounds`.

[5]The settings of adversarial training follows the experiments in `https://github.com/JiancongXiao/Adversarial-Rademacher-Complexity`.

