# OpenReview forum: "PAC-Bayesian Spectrally-Normalized Bounds for Adversarially Robust Generalization"
_NeurIPS.cc/2023/Conference — NeurIPS 2023 poster_

### Official Review · Reviewer_Ccn3 · 2023-06-09

**Soundness:** 2 fair
**Presentation:** 2 fair
**Contribution:** 2 fair
**Rating:** 4
**Confidence:** 4

**Summary:**

This work tries to develop a PAC-Bayesian spectrally-normalized robust generalization bound.

**Strengths:**

This work tries to understand robustness from theoretical perspective.

**Weaknesses:**

1. unclear definitions:
second line in Eq. (4)
$\mathbf{x}(\mathbf{w})=\arg \inf_{\left\|\mathbf{x}-\mathbf{x}^{\prime}\right\| \leq \epsilon} g_{\mathbf{w}}(\mathbf{x})$ (I guess it's $\mathbf{x}(\mathbf{w})=\arg \inf_{\left\|\mathbf{x}-\mathbf{x}^{\prime}\right\| \leq \epsilon} g_{\mathbf{w}}(\mathbf{x'})$)
$\mathbf{x}(\mathbf{w}')=\arg \inf_{\left\|\mathbf{x}-\mathbf{x}^{\prime}\right\| \leq \epsilon} g_{\mathbf{w}'}(\mathbf{x})$
$\inf_{\left\|\mathbf{x}-\mathbf{x}^{\prime}\right\| \leq \epsilon} g_{\mathbf{w}}(\mathbf{x})$
etc

2. Assumptions are not clearly described before theorem 1:
What are the conditions on $B$ and $\gamma$ for your theorem?
E.g., Neyshabur et al. (2017b) assumes $\mathcal{X}_{B, n}=[ \mathbf{x} \in \mathbb{R}^n \quad | \quad ||\mathbf{x}||_2 \leq B ]$  and  $\mathbb{P}_\mathbf{u} [\max_\mathbf{x} | f_\mathbf{w+u}(\mathbf{x})-f_\mathbf{w}(\mathbf{x}) |_\infty<\frac{\gamma}{4} ] \geq \frac{1}{2}$ (or $\ell_2$ norm).

3. Miss citation for Line 482.

4. What is $||W_i-W_i||$?

5. Please provide the details from Eq. (10) to Eq. (11).

6. As claimed in the contribution: **without any additional assumption**, **as tight as**, **how to obtain a robust generalization bound**.
My concerns are: **Does this bound provide new information for us?**  In my opinion, for (middle or large) DNNs, PAC-Bayes is not a tight bound but may provide inspiration for us to get a better generalization model. For the bound of this work, it just replaces $B$ with $B+\epsilon$. To me, it simply implies that $||\mathbf{x}||_2\le B$ (clean data), $||\delta||_2\le \epsilon$ (attack radius) thus $||\mathbf{x}+\delta||_2\le B+\epsilon$ (adversarial data), but it holds no significant meaning.

For the above reasons, I think this work should be rejected.


**Questions:**

ref Weaknesses

**Limitations:**

ref Weaknesses

---

> ### Author Rebuttal · Authors · 2023-08-03
>
> We thanks Reviewer Ccn3 for the comments and questions.
> ___
> **Q1, Q3 and Q4.** Unclear definition, what is $W_i-W_i$. Miss citation.
>
> A: Thanks for pointing out the typo and missing citation. Some prime’ are missed due to the full/half width issue. We fixed the typo in the updated version. In $W_i-W_i’$, the second one should be $W_i’$. The second line in Eq. (4) is equal to the first line. Line 482 is the tail bound [1].
>
> [1] Joel A Tropp. User-friendly tail bounds for sums of random matrices. Foundations of computational mathematics, 12(4):389–434, 2012.
> ___
> **Q2:** What are the conditions on B and γ for your theorem?
>
> A. Thanks for the question. B and γ are well-defined before the Theorem and it is the same as that in Neyshabur et al., 2017.
>
> B is first defined in line 65. It is the magnitude of the training sample. We will also add $||x||_2\leq B$ for all training samples $x$ in line 135.
>
> $\gamma$ is defined in line 147. It is the margin of the function value between true label and the max of false label.
> It is important to note that $\gamma$ is also the margin in Neyshabur et al., 2017. There exists a misunderstanding that $P_{u}\left[\max_{x \in X} |f_{w+u}(x)-f_{w}(x)|_\infty <\frac{\gamma}{4} \right]\geq \frac{1}{2}$ is not the assumption of $\gamma$. This inequality is the condition for Lemma 4, and it only serves as a bridge to prove the generalization bound.
>
> ___
> **Q5.** Please provide the details from Eq. (10) to Eq. (11).
>
> A. Thanks for the question. The first inequality is Lemma 6.2 (Eq. (9)). The second inequality is the tail bound (line 482).
>
> Q1-5 is about the typos or further detials. We will fixed it in the updated version. Below we answer your main question.
> ___
> **Q6:** Does this bound provide new information for us?
>
> **A:** Our main result offers **valuable and new information** into adversarially robust generalization. It is discussed in Section 5. We understand the main concern or confusion come from the fact that we did not emphasize it is new. We will make necessary revision to distinguish old and new information.
>
> The answer is provided above in common question 2. We hope it adequately addresses your concerns.
>
> We hope you also review our response to "common question 1" for additional context. Understanding the historical background of the theory of norm-based complexity is crucial in comprehending the significance of our findings. Further details will be provided below. Following your thought, we decomposed the question into two questions:
>
> 1) the significant meaning of the bounds and
>
> 2) the inspiration to get better model.
>
> ___
> **Q6.1.** the **significant meaning** of the bounds.
>
> A. replacing B by B + ε has its natural meaning.
>
> In standard generalization bound: standard generalization ≤ B × spectral complexity means that **standard generalization** is related to **clean samples** and spectral norm (of DNNs).
>
> In robust generalization bound: robust generalization ≤ (B + ε) × spectral complexity means that **adversarially robust generalization** is related to **adversarial examples** and spectral norm (of DNNs).
>
> Therefore, replacing B by B + ε has a natural meaning: it replaced clean samples by adversarial examples from standard to robust generalization.
>
> The important message of the bounds lies in the spectral complexity, rather than the training samples.
> ___
> **Q6.2.** The **inspiration** to provide better model.
>
> As you mentioned, the bound may provide inspiration for us to get a better generalization model. Let us discuss about the inspiration.
>
> **Firstly, why standard generalization bound is important?**
>
> It is proved that **standard generalization ≤ B × spectral complexity**. While the term "B" may not be directly controllable, the spectral complexity is within our control. Therefore, focusing on the controllable factor of spectral complexity presents a potential avenue to enhance generalization performance.
>
> **Secondly, why robust generalization bound is important?**
>
> It lies in the widely observed phenomenon where deep neural networks (DNNs) exhibit strong standard generalization but poor robust generalization. It is important to see what factors contribute to this difference. Then, the mismatch factors between standard and robust bound provide rich information to understand the disparity between standard and robust generalization.
>
> **Previous result** showed that **robust generalization≤ (B + ε) × spectral complexity and other factors**. $B+\epsilon$ (the adversarial example) is not controllable but the other factors are controllable. Therefore, spectral complexity and other factors (width, gradient, addition assumptions) are all potential factors to improve robustness.
>
> However, it is hard to verify which factors are more important, since we don’t know whether these bounds are tight in terms of additional factors, and whether these factors can be further reduced. Therefore, providing a better bound is an important problem in learning theory. Our paper give an affirmative answer.
>
> **We prove that robust generalization ≤ (B + ε) × spectral complexity.**
>
> Therefore, we dismiss the possibility of other factors (width, gradient, assumptions). At least from theory perspective, these other factors are solely comes from mathematic issue. It inspires engineers to pay more attention to spectral complexity as a potential factor to improve robustness. It is crucial to emphasize that the inspiration discussed is solely derived from our result.
>
> We hope our explanation can help understand why our result is important from an inspiration perspective.
>
> ___
> Overall, we believe our work provide a fundamental result to the learning theory comunity. Currently, the score "Soundness: 1 poor" and "Contribution: 1 poor" is very unfair to our work. We hope our answer can address your concern. We hope you can reconsider the significance of our work. If you have any further inquiries, please feel free to ask.

---

> > ### Comment · Reviewer_Ccn3 · 2023-08-15
> > **response**
> >
> > I thank authors for their careful reply.
> > As the typos in the equation (definition) destroyed my patience to carefully review this paper, I only spent an hour checking most mathematical parts.
> > I have to say, the current version is very unfriendly to a general reader in the community.
> >
> > All in all, I think the current manuscript has its merits but also obvious flaws, I sit on the fence for this paper.

---

> > > ### Author Response · Authors · 2023-08-15
> > > **Thanks for the response**
> > >
> > > Thank you very much for the response.
> > >
> > > 1) About the typos.
> > >
> > > Thanks for pointing out the typos about 1) missing prime in f(x) and Wi and 2) missing "=" in Equation (4). We have carefully proofread the paper to fixed other potential typos.
> > >
> > > 2) the current version is very unfriendly to a general reader.
> > >
> > > Thanks for pointing it out. Now, we have updated the paper based on the comments and questions for general readers.

---

### Official Review · Reviewer_FsEL · 2023-06-24

**Soundness:** 4 excellent
**Presentation:** 3 good
**Contribution:** 3 good
**Rating:** 7
**Confidence:** 5

**Summary:**

This paper improves previous PAC-Bayesian bounds on robust generalization. The previous bound in Farnia et al. (2018) has a term that is not bounded, and this work provides a bound to that term using the Lipschitzness of feed-forward ReLU networks. The basic idea is that coordinate-wise Lipschitzness preserves under L-inf perturbation.

**Strengths:**

I am quite familiar with this field and spent 3 hours reviewing this paper. Though I do not closely follow the robust generalization line of research, I believe that this work could be helpful to people working in the same direction. The overall framework is very clear and the results are intuitive and easy to understand. Though there are some confusing parts in some sections and the writing can still be improved, overall this looks like a nice paper and should make it to NeurIPS.

The overall framework is clean, and the proofs are easy to read. I quickly checked all the proofs, and they look good to me. I am not 100% sure that all proofs are sound, but even if there are small errors, they should be fixable because the results are very intuitive.

**Weaknesses:**

My only concern is that the significance of this work might not be obvious to a person who is not very familiar with this field. This work is addressing a very specific issue in a previous theorem, and while this issue is important, I think the authors should clarify more about why it is important, what is the main challenge and how this paper fixes it, right at the beginning of the paper. Right now I would say that those things are quite scattered. For example, the main challenge is in Section 6.2. I think the authors could include the following in the intro:
- What is the main issue in the previous PAC-Bayes bounds?
- Why is it difficult to fix this issue?
- What additional assumptions does this work make in order to fix this issue?
- What is the main technical contribution in the proof of this work?

As someone very familiar with learning theory and adversarial robustness, I can find the answers to these questions easily in the paper. However, if the authors want to appeal this work to a more general audience, I suggest them rearrange the paper a little bit, and discuss these questions in the intro.

**Questions:**

See above.

**Post rebuttal note:** I have read the rebuttal and other reviews. I prefer to keep my score.

**Limitations:**

Limitations are not discussed.

---

> ### Author Rebuttal · Authors · 2023-08-09
>
> We thanks Reviewer FsEL for the comments and questions.
>
> **Comment 1.** My only concern is that the significance of this work might not be obvious to a person who is not very familiar with this field.
>
> A. Thanks for the suggestions. We understand that the significance of this work might not be obvious to general readers. We will clarify
>
> 1)	Why the targeted problem is an important unsolved problem? And
>
> 2)	Why our result is significantly important?
>
> in Intro. Such discussion is also provided in our answer to common question 1.
>
>
> **Detailed suggestion.** I think the authors could include the following in the intro.
>
> A. Thanks for the suggestions. Some of them (2,3) are already provided in intro. The others (1,4) are discussed but not emphasized. We provided our detailed answer and detailed modification below.
>
> **S1.** What is the main issue in the previous PAC-Bayes bounds?
>
> This in discussed in line 55-56. We will add: “The inclusion of these supplementary assumptions and an additional term is less than ideal. These adjustments were made as a compromise to address mathematical complexities."
>
> **S2.** Why is it difficult to fix this issue?
>
> This is discussed line 48-51.
>
> **S3.** What additional assumptions does this work make in order to fix this issue?
>
> This is discussed in line 58.
>
> **S4.** What is the main technical contribution in the proof of this work?
>
> The technical contribution is detailed in the paragraph titled "Technical Contribution" on line 73. The current exposition is presented at a high level; we will now provide a more specific breakdown to the updated version:
>
> Our approach to solving this problem involves two main aspects:
>
> 1) We introduce a crucial inequality to address this problem, which is the preservation of weight perturbation bound under lp attack.
>
> 2) We restructure the proof by Neyshabur et al., in terms of the margin operator. This modification enables the application of the aforementioned inequality.

---

> > ### Comment · Reviewer_FsEL · 2023-08-13
> >
> > Thank the authors for the rebuttal. I have read the rebuttal and will keep my rating.

---

> > > ### Author Response · Authors · 2023-08-15
> > > **Thanks for the response.**
> > >
> > > Thanks for the response. We have updated the paper based on the comments and questions.

---

### Official Review · Reviewer_AB19 · 2023-07-07

**Soundness:** 3 good
**Presentation:** 3 good
**Contribution:** 2 fair
**Rating:** 6
**Confidence:** 3

**Summary:**

In this work, authors use PAC-Bayesian bound to characterize the generalization gap of adversarial robustness. Their work is mostly based on the bound derived from (Neyshabur et al., 2017b) so the resulting bound is valid for a deterministic model.


**Strengths:**

The major contribution from this work is the new PAC-Bayesian bound for adversarial robustness. The bound works for both Lp and non-Lp cases on both feed-forward networks and ResNets. Authors also compare their bounds with existing ones that only target specific attacks, and show their bounds are more generic. I appreciate the paper’s contribution of the bound (but I do have a question regarding the tightness of the bound which I will elaborate in the next part). The paper has pushed the use of PAC-Bayesian theories to scenarios beyond standard generalization.


**Weaknesses:**

The first weakness of the paper is probably that this newly derived bound largely relies on the work from Neyshabur et al., which has been phrased as an advantage of Theorem 1 being tight at least as Neyshabur et al.’s. However, this might be misleading as I believe the correct description would be when $\epsilon=0$ Theorem 1 reduces to Theorem 2 so they are as tight as each other. I am not sure I understand what does it mean for Theorem 1 and 2 to be equally tight when $\epsilon > 0$? Do you mean that what is in the big O notation is around the same magnitude? However, I do not see the value of $p$ (in $\ell_p$) plays any role in the new bound and the paper talks about the general $\ell_p$ robustness, thus I assume the bound works in any $\ell_p$ space. I hereby have the following question, if the bound is p–norm-agnostic, how come it is equally tight for all $p$? Does this bound simply characterize the robustness of the model in the largest ($\ell_\infty$) perturbation ball for a given $\epsilon$? If that is the case, perhaps the derived bound is pretty loose for $\ell_2$ perturbations. Can you elaborate more here (and in the paper) about the tightness of the bound for different $p$.

Another weakness is the empirical study to demonstrate the tightness of the bound in Table 1. This table is poorly captioned because I do not know how these models are trained, what architectures are used, what $\epsilon$ and attack techniques are used to report the numbers. Also, what does $\infty$ means for Theorem 3? Does this mean the assumption about gradient norm in Theorem 3 does not hold? Do you have experiments about the gradient norms? Moreover, I think it might be useful to use more models and more statistically significant numbers to show the proposed bound is tighter, like the measurement done in this paper [1].
[1] Jiang, Y., Neyshabur, B., Mobahi, H., Krishnan, D., & Bengio, S. (2019). Fantastic Generalization Measures and Where to Find Them. ArXiv, abs/1912.02178.


**Questions:**

I do not have further questions. I think the theoretical contribution is incremental so it may further increase my score if more empirical evaluations are conducted.

**Limitations:**

It might be nice to include such a paragraph.

---

> ### Author Rebuttal · Authors · 2023-08-03
>
> We thanks Reviewer AB19 for the comments and questions.
>
> **Comment.**  I think the theoretical contribution is incremental.
>
> A. Thanks for the comment, we will first answer this comment in the beginning. The technical novelty of our research goes beyond mere improvement. The **technical novelty is the most important part** in our work. Let us clarify below.
>
> Firstly, we provided a detailed comparison with Farnia et al. (2018) only because we both employ the Pac-bayes framework. This comparison aims to shed light on the mathematical difficulties involved in deriving robust bounds. However, it should not be mistaken as implying that our results are merely incremental improvements on Farnia et al. (2018). The discussion of Farnia et al. (2018) aims to convey two messages:
>
> 1) Existing technique cannot provide a robust bound, unless new assumption is introduced.
>
> 2) New mathematical tools is needed to prove a robust generalization bound.
>
> We understand our comparison to Farnia et al. (2018) mislead the reviewers to consider our work is incremental. We will make revision to address this concern. Please see our detailed answer to common question 1.
> ___
> **Q1.** The first weakness of the paper is probably that this newly derived bound largely relies on the work from Neyshabur et al., which has been phrased as an advantage of Theorem 1 being tight at least as Neyshabur et al.’s.
>
> A. Thanks for the question. I consider this aspect to be a strength rather than a weakness. It is essential to note that this particular question is connected to "common question 1," and we believe that the answer provided in our response to "common question 1" adequately addresses your concern.
> In summary, the bound presented by Neyshabur et al. holds significant value as a benchmark bound. Our findings successfully resolve the question that has spanned several years, namely, the tightness of a robust generalization bound.
> ___
> **Q2.** However, this might be misleading as I believe the correct description would be when ϵ=0 Theorem 1 reduces to Theorem 2 so they are as tight as each other. I am not sure I understand what does it mean for Theorem 1 and 2 to be equally tight when ϵ>0?
>
> A. Thank you for your question. Understanding the concept of "equally tight" can be viewed from two perspectives:
>
> At the first level, you are correct that as ε approaches 0, Theorem 1 converges to Theorem 2. Consequently, the two bounds become equally tight in this scenario.
>
> At the second level, when ε is greater than 0, it implies two things:
>
> 1) The term "B" (clean sample) in the standard bound corresponds to "B + ε" (adversarial example) in the robust bound.
>
> 2) The other factors in both bounds remain the same.
>
> To elaborate further, it is proved that "B + ε" exists in the lower bound, as demonstrated in Theorem 3 by Xiao et al. in 2022. As a result, when deriving the robust bound, we should replace "B" with "B + ε". Moreover, since the other factors remain unchanged in both bounds, this is why Theorem 1 and Theorem 2 are deemed equally tight.
>
> Xiao, J., Fan, Y., Sun, R., and Luo, Z.-Q. Adversarial rademacher complexity of deep neural networks. arXiv preprint arXiv:2211.14966, 2022.
> ___
> **Q3.** Do you mean that what is in the big O notation is around the same magnitude?
>
> A. No, it is not what we mean. Even though the answer is provided above, it is also important to notice that the big O notation omit a number 42 for both bounds.
> ___
> **Q4:** However, I do not see the value of p (in ℓp) plays any role in the new bound and the paper talks about the general ℓp robustness, thus I assume the bound works in any ℓp space.
>
> A. Thanks for the question, the main result (Theorem 1) is stated in ℓ2 norm. It is easy to calibrate the results to other ℓp attack by a factor $n^{1/2-1/p}$, where $n$ is the dimension of $x$. We delete this discussion when we condensed the article to 9 pages. Thanks for pointing it out, and we realize that deleting this part will cause unnecessary confusion. We will add the discussion to the updated version.
> ___
> **Q5.** Another weakness is the empirical study to demonstrate the tightness of the bound in Table 1.
>
> A. Thanks for the questions. There exists a misunderstanding of Table 1. Table 1 aims to show the effect of spectral complexity on standard and robust generalization. The discussion about Table 1 is provided in Sec. 5.
>
> Table 1 is not to prove the tightness of the bound. Actually, we don’t need any experiments to show the proposed bound is tighter. The proposed bound is strictly tighter since we reduce an additional positive term.
>
> We hope the fact above can address your main concern. The following detail questions are about fair comparison, which is not related to the main concern if our answer above is helpful. Yet we can still provide our answer.
>
> **Q5.1.** Training details
>
> As we state in Line 214, the experiment follows [3]. We use VGG-19 networks. As for adversarial training, we use a standard PGD-20 attacks with $\epsilon=0.5$.
>
> [3] Neyshabur, B., Bhojanapalli, S., McAllester, D., and Srebro, N. Exploring generalization in deep learning. arXiv preprint arXiv:1706.08947, 2017a.
>
> **Q5.2.** Also, what does ∞ means for Theorem 3? Does this mean the assumption about gradient norm in Theorem 3 does not hold? Do you have experiments about the gradient norms?
>
> ∞ means the bound is unbounded without additional assumption, as we state in line 198. Actually, it is highly non-trivial to estimate the lower bound of the gradient norms over the whole domain. We would like to replace ∞ by N/A in Table 1.
>
> **Q5.3.** Moreover, I think it might be useful to use more models and more statistically significant numbers to show the proposed bound is tighter.
>
> We don’t need any experiments to show the proposed bound is tighter. In mathematics, the proposed bound is strictly tighter.
> ___
> We hope our responses have addressed your questions adequately. If you have any further inquiries, please feel free to ask.

---

> ### Comment · Reviewer_AB19 · 2023-08-14
>
> Thanks for addressing (most of) my concerns. I have increased my score to 6.

---

> > ### Author Response · Authors · 2023-08-15
> > **Thanks for the response.**
> >
> > Thanks for the response. We have updated the paper based on the comments and questions.

---

### Official Review · Reviewer_nNum · 2023-07-07

**Soundness:** 3 good
**Presentation:** 3 good
**Contribution:** 2 fair
**Rating:** 5
**Confidence:** 4

**Summary:**

This paper provides a tighter bounds for robust generalization compared to previous results and as tight as standard generalization.

**Strengths:**

1. The paper studies an important topic on adversarial robustness and provide a tighter bound with detailed theoretical analysis.
2. The paper is well-written and the conclusions are sound as far as I understand (I didn't check the proof in Appendix).

**Weaknesses:**

* As the paper is an improvement on Farnia et al. (2018), thus the technical novelty is less significant.
* The authors should explain why a tighter upper bound is insightful for the robustness community. In my understanding, the robust generalization bound is usually used to explain which factor affects the robust generalization, or why robust generalization is worse than standard generalization. The authors should discuss more on the implications from the new result.

**Questions:**

Please see the weaknesses part.

**Limitations:**

There is no discussion on limitations.

---

> ### Author Rebuttal · Authors · 2023-08-02
>
> We thanks Reviewer nNum for the comments and questions.
> ___
> **Q1:** As the paper is an improvement on Farnia et al. (2018), thus the technical novelty is less significant.
>
> **A:** Thanks for the question. the technical novelty of our research goes beyond mere improvement. The **technical novelty** is the **most important** part in our work.
>
> Let us clarify below.
>
> Firstly, we provided a detailed comparison with Farnia et al. (2018) only because we both employ the Pac-bayes approach. This comparison aims to shed light on the mathematical difficulties involved in deriving robust bounds. However, it should not be mistaken as implying that our results are merely incremental improvements on Farnia et al. (2018). The discussion of Farnia et al. (2018) aims to convey two messages:
>
> 1) Existing technique cannot provide a robust bound, unless new assumption is introduced.
>
> 2) New mathematical tools is needed to prove a robust generalization bound.
>
> Secondly,  as we discussed in line 117-130, related work in this field (no less than 7 papers, including Farnia et al.,'s Pac-Bayes approach) tried to extend the norm-based bounds to robust settings. However, the main difficulty is mathematics. The inclusion of supplementary assumptions or additional term presented in these work is less than ideal. These adjustments were made as a compromise due to mathematical issue. However, an ideal bound should be a clean extension of the generalization bound in standard settings.
>
> Therefore, the technique to provide such an extension is important to the community. The significance of our paper lies in providing the math technique to resolve this problem.
>
> Finally, since this is a common question, we provide our answer regarding the **significance of our result** in more detailed in “common question 1”. We understand our comparison to Farnia et al. (2018) mislead the reviewers to consider our work is incremental. We will make revision to address this concern.
>
> We believe our detailed answer in "common question 1" could address your concern.
>
> ---
> **Q2:** The authors should explain why a tighter upper bound is insightful for the robustness community.
>
> **A:** Thanks for the question. Our bound offers **significant new insights** into adversarial robustness. It is discussed in Section 5. We understand the main concern or confusion come from the fact that we did not emphasize it is new. We will make necessary revision to distinguish old and new information.
>
> As this is a common question from reviewers, we discuss the new insight of our result above in “common question 2”.
>
> ___
> We hope our responses have addressed your questions adequately. If you have any further inquiries, please feel free to ask.

---

> > ### Comment · Reviewer_nNum · 2023-08-14
> > **Post Rebuttal**
> >
> > Thanks for the response.
> > I have read the response. The response looks good but I believe that the authors still need some revision for the paper to make the presentation clearer. I'll keep my rating.

---

> > > ### Author Response · Authors · 2023-08-15
> > > **Thanks for the response.**
> > >
> > > Thanks for the response. We have updated the paper based on the comments and questions.

---

### Author Rebuttal · Authors · 2023-08-02

We thanks all the Reviewers for the comments and questions. We will first answer three common questions.

**Common Question 1:** Significance of the result. (Reviewer nNum & AB19)

**A:** Our finding is not just an incremental improvement on Farnia et al. (2018); it holds **significant importance** in the field of learning theory, as highlighted by Reviewer FsEL. Let us clarify below.

Short version:

There's an important unsolved question over the past few years: Is there a clean/tight extension of norm-based generalization bound to robust settings?

No less than 7 papers (The work discussed in line 117 to 130) aimed at solving this problem. However, they have faced mathematical challenges. The most important contribution of our paper is to provide the **mathematical technique** (Presented in Sec. 6) to resolve this problem.

We understand our comparison to Farnia et al. (2018) **mislead the reviewers to consider our work is incremental.** However, this comparison aims to shed light on the mathematical difficulties involved in deriving robust bounds. We will make revision to address this concern.
___
Long version:

**1) Importance benchmark of norm-based generalization bounds.**

Firstly, norm-based complexity stands as a crucial generalization measure for ML models. This field has seen significant interest, with numerous published papers (not less than 30) exploring this topic. In 2017, Neyshabur et al. introduced a spectral-normalized bound for standard settings, using Pac-Bayes approaches. After that, progress in the field of norm-based bounds for standard training has experienced a temporary stall in recent years, with no tighter bounds being proposed.

(Additionally, similar spectral norm bound is provided by Bartlett et al., 2017 using covering number approach. The above-mentioned two results hold fundamental significance and have been widely discussed in prominent ML courses, such as Stanford CS229M (Lecture 5) and MIT Statistical Learning Theory (Lecture 16).)

As such, Neyshabur et al.,'s (or Bartlett et al.,'s) bounds serve as essential 'benchmark' generalization bounds. Therefore, we refer to this bound as 'benchmark bound'. A direct question arises.


**2) Fundamental question: Is there a clean/tight extension of norm-based bound to robust settings?**

However, this question troubles the community in the last few years.

Researchers found that it is mathematically challenging to derive robust bounds using both Pac-bayes approach (Farnia et al., 2018) and covering number approaches (the work discussed in lines 117-130).

As a result, these work often consider simplifying cases or introducing additional assumptions to bypass the difficulty. The resulting robust bounds is much larger than the standard bound, exhibit higher dependence on factors, such as depth and width, or include additional terms. It remains uncertain whether further reduction is possible. Such attempts did not offer a clean bound for robust generalization.

The significance of our paper lies in providing the math technique to resolve this problem. We demonstrate that the robust generalization bound can achieve the benchmark bound. All the higher dependencies on factors, additional terms, or extra assumptions can be eliminated.

**3) Our paper resolve this question.**

More importantly, our results temporarily close this question, without a tighter (norm-based) bound for standard generalization, no tighter (norm-based) bound for robust generalization will be provided.

___
**Common Question 2:** New insight to the robustness community. (Reviewer nNum & Ccn3)

**A: Our bound offers significant new insights into adversarial robustness**, as discussed in Sec. 5. One main source of confusion stems from the fact that we did not emphasize this crucial information in Sec. 5 cannot be provided by previous research.  We will make necessary revision to distinguish old and new information.

Let us explain it below.

It is widely recognized that deep neural networks (DNNs) demonstrate good standard generalization ability but often exhibit poor robust generalization. A key question that needs to be addressed is:

**What factors contribute to such a significant difference?** (Line 204)

Previous studies have shown that the robust bound is much larger than the standard bound. Therefore, the mismatch term between these two bounds might be a contributing factor to the significant difference. For instance, the robust bound might have a higher dependence on width or include an additional term related to gradient information. Such factors (width, gradient) could potentially explain the disparity between standard and robust generalization.

**Current Hypothesis: The significant difference is due to Robust Bound >> Std Bound.**

However, verifying this hypothesis has been challenging because it remains unclear whether the existence of these factors is due to mathematical issues.

Our results provide a definitive answer and dismiss this possibility. In essence,

**Our result: Robust Bound $\approx$ Std Bound.**

We have shown that the additional factors (dependence, additional terms, assumptions) are solely due to mathematical considerations. Consequently, from a norm-based complexity perspective, they do not contribute to the significant disparity.

Therefore, in Section 5, we focus on attack intensity and spectral complexity.

From the norm-based complexity perspective, we show that only **attack intensity** and **spectral complexity**, (but not other factors), contribute to the significant disparity. We believe this is a significant new insights into adversarial robustness.

___

**Common Question 3: Limitation.**

We provide the limitation below, which will be added to Sec. Conclusion.

The main limitation is that the norm-based bounds are all large in practice, as it is shown in Table 1. How to obtain smaller bound in a practical scenario is an open problem.

---

### Decision · Program_Chairs · 2023-09-21

**Decision:**

Accept (poster)

**Comment:**

The reviewers appreciated the improvements in the bound compared to previous work.
The main contribution compared to Farnia '18 and Neyshabur '17 is that the authors use a different perturbation quantity (perturbation of margin) for robust compared to standard generalization that allows tighter bounds with fewer assumptions, based on the Pac-Bayes framework (as in previous works) - the writing is also quite clear in pointing this out. As usual, it's unclear if these bounds are useful in practice, but that is a criticism that applies to deep learning theory in general...